# Acceleration in Distributed Sparse Regression

**Marie Maros**
School of Industrial Engineering
Purdue University
mmaros@purdue.edu

**Gesualdo Scutari**
School of Industrial Engineering
Purdue University
gscutari@purdue.edu

## Abstract

We study acceleration for distributed sparse regression in *high-dimensions*, which allows the parameter size to exceed and grow faster than the sample size. When applicable, existing distributed algorithms employing acceleration perform poorly in this setting, theoretically and numerically. We propose an accelerated distributed algorithm suitable for high-dimensions. The method couples a suitable instance of accelerated Nesterov's proximal gradient with consensus and gradient-tracking mechanisms, aiming at estimating locally the gradient of the empirical loss while enforcing agreement on the local estimates. Under standard assumptions on the statistical model and tuning parameters, the proposed method is proved to globally converge at *linear* rate to an estimate that is within the *statistical precision* of the model. The iteration (i.e., gradient oracle) complexity scales as $\mathcal{O}(\sqrt{\kappa})$, while the communications per iteration are at most $\widetilde{\mathcal{O}}(\log m/(1-\rho))$, where $\kappa$ is the restricted condition number of the empirical loss, $m$ is the number of agents, and $\rho \in [0,1)$ measures the network connectivity. As by-product of our design, we also report an accelerated method for high-dimensional estimation over master-worker architectures, which is of independent interest and compares favorably with existing works.

## 1 Introduction

Consider a multiagent system composed of $m$ machines, connected throughout a fixed, undirected graph–we will refer to such networks as *mesh* networks, in contrast to centralized topologies such as master-worker architectures, where there exists a node connected to all the others. The ultimate goal is to solve the stochastic (quadratic) optimization (learning) problem

$$\theta^\star \in \underset{\theta \in \mathbb{R}^d}{\operatorname{argmin}} \left\{ \bar{f}(\theta) \triangleq \frac{1}{2}\mathbb{E}_{(x,y)\sim\mathbb{P}}[(x^\top \theta - y)^2] \right\}. \tag{1}$$

Each of the $m$ agents has access to $n$ i.i.d observations, drawn from the unknown, common distribution $\mathbb{P}$ on $\mathcal{Z} \subseteq \mathbb{R}^p$, and collected in the set $\mathcal{S}_i$. The associated local empirical estimate of $\bar{f}$ reads

$$f_i(\theta) \triangleq \frac{1}{2n} \sum_{j \in \mathcal{S}_i} (x_j^\top \theta - y_j)^2, \tag{2}$$

where $x_j \in \mathbb{R}^d$ is the vector of predictors and $y_j \in \mathbb{R}$ the associated response. The overall Empirical Risk Minimization (ERM) over the network is then

$$\widehat{\theta} \in \underset{\theta \in \Omega : \mathcal{R}(\theta) \leq R}{\operatorname{argmin}} \left\{ f(\theta) \triangleq \frac{1}{m} \sum_{i=1}^m f_i(\theta) \right\}, \tag{3}$$

where $\mathcal{R}$ is a (convex) regularizer (with $R > 0$) and $\Omega$ is some (convex) subset of $\mathbb{R}^d$.

36th Conference on Neural Information Processing Systems (NeurIPS 2022).

We study estimators under *high-dimensional scaling*, meaning that the ambient dimension $d$ exceeds (and grows faster than) the total sample size $N = m \cdot n$. The desired parameter $\theta^\star$ is assumed to lie in a smaller subset of $\Omega$ or is well approximated by a member of it–such structural constraints are enforced by the regularizer $\mathcal{R}$ in (3). Examples include sparse linear models with $\ell_1$ regularizers, low-rank matrix recovery via nuclear norm, and matrix regression with soft-rank constraints [1].

The goal is to compute an estimate of $\theta^\star$ by solving the ERM (3) that is within the statistical error of the model $\|\widehat{\theta} - \theta^\star\|$. We target first-order methods. Despite the vast literature of distributed optimization (see Sec. 1.1), only very recently such guarantees have been established on mesh networks for the LASSO problem in the projected form [26] [as (3)] and in the Lagrangian form [11]. The benchmark is the scheme in [26], termed NetLASSO: under restricted notions of strong convexity and smoothness of $f$ (see Sec. 2)–which hold with high probability for a variety of data generation models–as well as $s \log d / N = o(1)$–a condition necessary for statistical consistency in $s$-spare linear regression–the iterates generated by NetLASSO reach an $\varepsilon$-neighborhood of a statistically optimal solution in

$$\widetilde{\mathcal{O}} \left( \kappa \frac{\log m}{1 - \rho} \log 1/\varepsilon \right) \quad \text{number of communications,} \tag{4}$$

where $\kappa \geq 1$ is the condition number of $f$ in (3) restricted to sparse directions, and $\rho \in [0, 1)$ measures the connectivity of the network–the smaller the more connected the graph ($\widetilde{\mathcal{O}}$ hides log-factors on optimization parameters). This shows linear rate up to statistical precision, whose dependence on $\kappa$ matches that of the centralized Projected Gradient Algorithm (PGA) [1].

Since the bottleneck in distributed systems is often represented by the cost of communications (in comparison with local computations) [6, 15], a key question is whether the communication complexity (4) is improvable while still preserving the same *statistical* guarantees and local computational complexity (first-order methods). This is particularly sensible when dealing with ill-conditioned population losses, resulting in very large values of $\kappa$. To improve the dependency on $\kappa$, a natural approach is to employ some form of acceleration. Despite the vast literature of accelerated methods in distributed optimization, this remains an open problem in *high-dimensions*, as elaborated next.

## 1.1 Distributed acceleration in high-dimensions: Challenges and open problems

Accelerating first-order methods over mesh networks (undirected graphs) has received attention in the last few years; distributed schemes in the primal domain include [22, 32, 33, 14, 12, 24] while [25, 28, 13] are accelerated dual or penalty-based methods. They are suitable to solve distributed, *unconstrained, smooth* optimization problems, with the exception of [33], which decentralizes the accelerated proximal-gradient method, and thus is applicable also to the constrained ERM (3).

Albeit different, when applied to the minimization of a $L_f$-smooth and $\mu_f$-strongly convex loss $f = 1/m \sum_{i=1}^{m} f_i$, with each $f_i$ being $L_i$-smooth and $\mu_i$-strongly convex, they all provably achieve linear convergence to the minimizer of $f$, some [25, 13, 14, 12, 28] with communication complexity scaling with $\sqrt{\kappa_\ell}$ ($\kappa_\ell = \max_i L_i / \min_i \mu_i$ is the local condition number) while others [32, 33, 24] with $\sqrt{\kappa_f}$ ($\kappa_f = L/\mu$ is the condition number of $f$). Note that in general $\kappa_f < \kappa_\ell$; hence, the latter group is preferable to the former. For weakly convex losses $f_i$'s, convergence is certified at *sublinear* rate, the optimality gap on the objective value vanishes as $\mathcal{O}(1/t^2)$, where $t$ is the iteration index.

When applied to the constrained ERM (3) in high-dimensions, these convergence results are unsatisfactory. **First**, they are of pure optimization type, and do not provide any statistical guarantee–computing below the statistical noise would result in a waste of resources. **Second**, for fixed $d$ and $N$, with $d > N$ (high-dimensions), they would certify sublinear convergence rates, as the empirical loss $f$ is convex but not strongly convex *globally*–the $d \times d$ Hessian matrix $\nabla^2 f$ has rank at most $N$. This provides pessimistic, non-informative predictions, which contrast with (4), proving instead that linear rates up to the statistical precision are achievable by nonaccelerated distributed methods. **Third**, under the scaling $d/N \to \infty$ ($d$ growing faster than $N$)–the typical regime in high-dimensions [29, 1]–convergence analyses in the aforementioned papers break down; they all require *global smoothness* of the local $f_i$'s and global $f$ losses, a property that does not hold when $d/N \to \infty$. If fact, for commonly used designs of predictors $x_i$'s, the Lipschitz constant of $\nabla f$ grows indefinitely with $d/N$ [29]. **Fourth**, if nevertheless simulated, the only distributed accelerated scheme applicable to the constrained ERM (3), that is DPAG [33], performs poorly in high-dimensions, as Fig. 1 shows. In the figure we plot the average

estimation error, defined as $(1/m) \sum_{i=1}^{m} \| \theta_i^t - \theta^\star \|^2$, versus the iterations indexed by $t$, where $\theta_i^t$ is the estimate of $\theta^\star$ from agent $i$ at iteration $t$ generated by DPAG solving a projected LASSO over a mesh network (a special instance of (3) with $\mathcal{R}(\theta) = \|\theta\|_1$), and $\theta^\star$ is the population $s$-sparse minimizer in (1). We contrast DPAG with the new distributed accelerated method proposed in this paper. As benchmark, we also plot the estimation error generated by NetLASSO (non-accelerated method) [26].

Despite the good performance of DPAG in low-dimensions (i.e., $N > d$) [33], the figure shows that acceleration as in DPAG is no longer effective in high-dimensions ($d > N$); DPAG achieves estimates of $\theta^\star$ that are worse than those achievable by non accelerated schemes, yielding estimation errors that are much larger than centralized statistical ones. Further, in our experiment we observed that this gap grows with $d/N$. In contrast, the proposed distributed acceleration exhibits linear convergence up to *centralized* statistical precision, at a rate faster than the nonaccelerated NetLASSO.

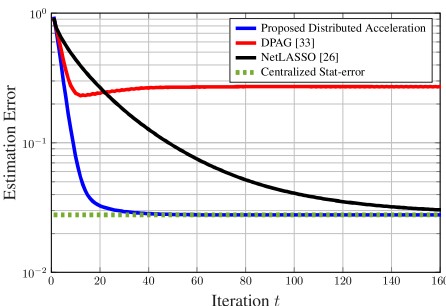

**Figure 1:** Projected LASSO: Average estimation error vs. iterations. Existing distributed accelerations (DPAG) may not work in high-dimension! The proposed accelerated method outperforms nonaccelerated distributed schemes (NetLASSO) while achieving centralized statistical accuracy. See Section A.2 for details

We conjecture that the unsatisfactory performance of DPAG [33] is due to the following facts: **(i)** the empirical loss of LASSO is strongly convex only over sparse vectors (see Sec. 2). Hence, fast convergence is expected only when traveling such a curved region of the landscape; **(ii)** the scheme [33] can be interpreted as a decentralization of the accelerated proximal gradient based on the Nesterov's first method [21]; as such, it does not implement any mechanism enforcing the trajectory at which gradients are evaluated to be (approximately) sparse. This suggests that acceleration in the distributed high-dimensional setting needs to be rethought–new designs and convergence analyses are needed. This is what this paper is about.

## 1.2 Main contributions

**Algorithm design:** We propose a distributed algorithm for high-dimensional estimation achieving acceleration provably. The method decentralizes Nesterov's accelerated proximal gradient (form III) [20] via consensus dynamics and a gradient tracking mechanism [7, 30, 19], the former enforcing an agreement on the agents' estimates and the latter aiming at tracking locally the gradient of $f$.

The rationale to hinge on Nesterov's third scheme rather than form I, as used instead so far in all accelerated distributed algorithms, is to enforce feasibility (with respect to the norm ball constraint $\mathcal{R}(\theta) \le R$) of the momentum-sequence generated by the algorithm, the one over which gradients are evaluated. For instance, when solving a constrained LASSO (i.e., $\mathcal{R}(\theta) = \|\theta\|_1$), this yields sequences that are approximately sparse, thus traveling the landscape of an "almost" strongly convex and smooth loss. In fact, our analysis shows that this is the key enabler to retain fast, linear convergence to statistically optimal solutions in high-dimension, where $f$ is no longer strongly convex globally.

**Statistical-computational guarantees:** Under standard notions of restricted strong convexity and smoothness of the empirical loss, which hold with high probability for a variety of statistical models, and proper algorithm tuning, the iterates generated by the proposed scheme converge at *linear* rate to a limit point that is within a fixed tolerance from the unknown $\theta^\star$. When customized to specific statistical models–including sparse linear models with $\ell_1$ regularization, low-rank matrix recovery with nuclear norm, and matrix completion–the tolerance becomes of the order of the statistical error $\|\hat{\theta} - \theta^\star\|$. Specifically, for such models and arbitrary network connectivity $\rho$, the iterates generated by the algorithm enter an $\varepsilon$-neighborhood of a *statistically optimal* estimate of $\theta^\star$ after

$$\mathcal{O}\left(\sqrt{\kappa} \log 1/\varepsilon\right) \quad \text{and} \quad \widetilde{\mathcal{O}}\left(\sqrt{\kappa} \, \frac{\log m}{1-\rho} \log 1/\varepsilon\right) \tag{5}$$

numbers of iterations (gradient evaluations) and communications, respectively, where $\widetilde{\mathcal{O}}$ hides log-factors on optimization parameters (recall $\kappa$ is the restricted condition number). This improves on the communication complexity (4) of NetLASSO [26] applied to $\ell_1$-constrained LASSO, showing the more favorable square-root scaling on $\kappa$, typical of accelerated methods in low-dimension. Furthermore, the rates in (5) are showed to be invariant under high-dimensional scaling $d/N \to \infty$, as long as the statistical error $\|\hat{\theta} - \theta^\star\|$ remains constant.

**A special case:** A by-product of the proposed method is its customization to master-worker architectures, which is of independent interest and compares favorable with existing accelerated centralized designs [16, 31], proposed for $\ell_1$-least squares (in Lagrangian form); they are *double-loop* methods, combining the accelerated proximal gradient method [27, 5, 20] with homotopy continuation.

## 2 Setup and Background

This section states the main assumptions on the ERM (3) and the network underlying our derivations.

● **RSC & RSM:** We anticipated that, in high-dimension, $f$ in (3) is not strongly convex uniformly (as $d > N$). However, strong convexity and smoothness hold along a restricted set of directions, for a variety of statistical estimations [29], which is enough to employ fast convergence and favorable statistical guarantees of the PGA [1]. Here we postulate the same properties for the landscape of $f$.

**Assumption 1** (Global RSC & RSM [1]). *The loss function $f$ in (3) satisfies*

*(i) Restricted Strong Convexity (RSC) with respect to $\mathcal{R}$ with (positive) parameters $(\mu/2, \tau_\mu)$ over $\Omega$:*

$$f(x) - f(y) - \langle \nabla f(y), x - y \rangle \geq \frac{\mu}{4}\|x - y\|_2^2 - \frac{\tau_\mu}{2}\mathcal{R}^2(x - y), \quad \forall x, y \in \Omega; \qquad (6)$$

*(ii) Restricted Smoothness (RSM) with respect to $\mathcal{R}$ with (positive) parameters $(2L, \tau_L)$ over $\Omega$:*

$$f(x) - f(y) - \langle \nabla f(y), x - y \rangle \leq L\|x - y\|_2^2 + \frac{\tau_L}{2}\mathcal{R}^2(x - y), \quad \forall x, y \in \Omega. \qquad (7)$$

*The restricted condition number is defined as $\kappa \triangleq L/\mu$ and it is assumed $\kappa \geq 1$.*

In the distributed setting, the RSM property is required to hold also locally.

**Assumption 2** (local RSM). *Each loss function $f_i$ satisfies local RSM with respect to $\mathcal{R}$ with (positive) parameters $(2\gamma_\ell, \tau_\ell)$ over $\Omega$:*

$$f_i(x) - f_i(y) - \langle \nabla f_i(y), x - y \rangle \leq \gamma_\ell\|x - y\|_2^2 + \frac{\tau_\ell}{2}\mathcal{R}^2(x - y), \quad \forall x, y \in \Omega; \qquad (8)$$

● **Decomposable regularizer $\mathcal{R}$:** This notion plays a key role for statistical consistency of $\hat{\theta}$ in high-dimension [1]. Decomposability is defined with respect to two subspaces of $\Omega$, namely: $\mathcal{M}$, known as the *parameter* subspace, capturing constraints from the model (e.g., vectors with a particular support or a subspace of low-rank matrices); and $\bar{\mathcal{M}}^\perp$, referred as *perturbation* subspace, representing the deviations from the model subspace $\mathcal{M}$. We also require Lipschitz continuity of $\mathcal{R}$ over $\bar{\mathcal{M}}$.

**Assumption 3.** *(i) The regularizer $\mathcal{R} : \Omega \to \mathbb{R}$ is a norm, with $\Omega \subseteq \mathbb{R}^d$ convex and $0 \in \Omega$ without loss of generality. Furthermore, given a subspace pair $(\mathcal{M}, \bar{\mathcal{M}}^\perp)$, such that $\mathcal{M} \subseteq \bar{\mathcal{M}}$:*

*(ii) $\mathcal{R}$ is $(\mathcal{M}, \bar{\mathcal{M}}^\perp)$-decomposable, that is, $\mathcal{R}(x + y) = \mathcal{R}(x) + \mathcal{R}(y)$, for all $x \in \mathcal{M}$ and $y \in \bar{\mathcal{M}}^\perp$;*

*(iii) When $\bar{\mathcal{M}} \neq \{0\}$, $\mathcal{R}$ is $\Psi(\bar{\mathcal{M}})$-Lipschitz over $\bar{\mathcal{M}}$ with respect to some norm $\| \bullet \|$: $\Psi(\bar{\mathcal{M}}) \triangleq \sup_{\theta \in \bar{\mathcal{M}} \setminus \{0\}} \mathcal{R}(\theta)/\|\theta\|$. If $\bar{\mathcal{M}} = \{0\}$, we set $\Psi(\{0\}) = 0$.*

Regularizers sharing the above properties over certain pairs $(\mathcal{M}, \bar{\mathcal{M}}^\perp)$ are ubiquitous in M-estimation problems [29]; examples include the $\ell_1$ norm, the (overlap) group norm, and the nuclear norm.

● **Statistical models:** RSC and RSM properties can be certified with high probability by a variety of random data generations. Here, we consider the following widely used statistical models, which cover a variety of estimation tasks, such as sparse linear models with $\ell_1$ regularization, low-rank matrix recovery with nuclear norm, and matrix regression with soft-rank constraints [17, 1, 29].

**Assumption 4.** *The random predictors $x_i \in \mathbb{R}^d$ are i.i.d. and fulfill one of the following conditions:*

    *(i) $x_i \sim \mathcal{N}(0, \Sigma)$ and are i.i.d., for some $\Sigma \succ 0$;*

    *(ii) $x_i$ are $(\tau^2, \Sigma)$−sub-Gaussian and i.i.d., with $\Sigma \succ 0$;*

    *(iii) $x_i = e_j$ where $j \sim uniform[1, d]$ and i.i.d.*

**Lemma 1.** *Under Assumption 4(i), there exist universal constants $c_0, c_1, c_2 > 0$ such that, with probability at least $1 - c_0 \exp(-c_1 N + \log m)$, $N \geq 10$, Assumptions 1 hold with parameters*

$$\mu = \lambda_{\min}(\Sigma), \quad \tau_\mu = c_2 \mathbb{E}[\mathcal{R}^*(x_i)]^2/N \quad and \quad L = \lambda_{\max}(\Sigma), \quad \tau_L = c_2 \mathbb{E}[\mathcal{R}^*(x_i)]^2/N,$$

*and so does Assumption 2 with parameters $\gamma_\ell = (2m + 1)\lambda_{\max}(\Sigma)$ and $\tau_\ell = c_2(\mathbb{E}[\mathcal{R}^*(x_i)]^2)/n$, where $\mathcal{R}^*$ is the dual norm of $\mathcal{R}$. When $\mathcal{R}(\cdot) = \|\cdot\|_1$, $(\mathbb{E}[\mathcal{R}^*(x_i)])^2 \leq 9(\max_j (\Sigma)_{jj}) \log(d)$.*

The proof of Lemma 1 follows from that of [1, Prop. 1]. Similar results under Assumptions 4(ii)-(iii) can be found in the supplementary material, see Appendix A.7.

• **On the network:** Agents are connected throughout a communication network, modelled as an undirected graph $\mathcal{G} = \{\mathcal{V}, \mathcal{E}\}$, where the vertices $\mathcal{V} = [m] \triangleq \{1, \ldots, m\}$ correspond to the agents and $\mathcal{E}$ is the set of edges of the graph; $(i, j) \in \mathcal{E}$ if and only if there is a communication link between agents $i$ and $j$. The set of neighbors of each agent $i$ is denoted by $\mathcal{N}_i \triangleq \{j \in \mathcal{V} : (i, j) \in \mathcal{E}\} \cup \{i\}$. In the proposed algorithms, agents will average neighbouring information using weights as below.

**Assumption 5** (Gossip matrices). *Define $W = [w_{ij}]_{ij=1}^m$; the following hold: **(i)** $W$ is $\mathcal{G}$-congruent, that is, $w_{ij} > 0$, if $(i, j) \in \mathcal{E}$; otherwise $w_{ij} = 0$; furthermore, $w_{ii} > 0$, for all $i \in [m]$; **(ii)** $W = W^\top$ and $W\mathbf{1} = \mathbf{1}$ (stochastic); and **(iii)** there holds $\rho \triangleq \|W - J\|_2 < 1$, where $J \triangleq \mathbf{1}\mathbf{1}^\top/m$.*

Assumption 5 is quite standard in the literature of distributed algorithms and is satisfied by several weight matrices; see, e.g., [18]. Note that $\rho < 1$ by construction, as long as $\mathcal{G}$ is connected. Roughly speaking, $\rho$ measures how fast the network mixes information; the smaller $\rho$, the faster the mixing.

## 3  Algorithm Design

At the core of our distributed accelerated algorithm there is an instance of Nesterov's acceleration proximal gradient method, which we identified to be suitable for high-dimensional estimation. We begin presenting this method in the centralized setting, along with its statistical guarantees (Sec. 3.1); we then proceed to design and analyze a suitable decentralization to mesh networks (Sec. 3.2).

### 3.1  Warm-up: A Nesterov's acceleration (form III) for high-dimensional estimation

Consider (3) in the centralized setting (e.g., master-worker systems). Assumption 1 shows that $f$ is strongly convex and smooth only along a restricted set of directions. To enable fast convergence, the algorithm design should produce trajectories (approximately) traveling such portions of the landscape of $f$. To handle this situation while achieving acceleration we propose to use Nesterov's method written in form III [9, Algorithm 11] equipped with the projected operator to enforce feasibility of all the sequences generated by the algorithm. This ensures that gradients are always evaluated at feasible points of (3), which we prove unlock linear convergence (up to a tolerance) at accelerated rate.

Given feasible $\theta^0$ and $v^0$, the main iterate $\{\theta^t, v^t, z^t\}$ of the algorithm reads: for any $t = 0, 1, \ldots,$

$$z^t = \frac{1}{\beta + 1}\theta^t + \frac{\beta}{\beta + 1}v^t,$$

$$v^{t+1} = \Pi_{\Omega \cap \mathcal{R}(v) \leq R}\left((1 - \beta)v^t + \beta z^t - \frac{\alpha}{\beta}\nabla f(z^t)\right); \tag{9}$$

$$\theta^{t+1} = \beta v^{t+1} + (1 - \beta)\theta^t.$$

where $\alpha \in (0, 1)$ and $\beta \in (0, 1)$ are parameters to tune, and $\Pi_{\Omega \cap \mathcal{R}(v) \leq R}(\bullet)$ denotes the Euclidean projection (of its argument) onto the convex set $\{\theta \in \Omega : \mathcal{R}(v) \leq R\}$; for specific estimators such as $\ell_1$-constrained or Lagrangian LASSO, this projection has a closed form expression [8, 11]. Notice that, under feasibility of $\theta^0$ and $v^0$, the sequences $\{\theta^t\}$, $\{v^t\}$ and $\{\theta^t\}$ are feasible, for all $t$.

A wide variety of accelerated methods exists in the centralized setting (see, e.g., [9] for a recent overview), and most variants settle the infeasibility of the $z^t$ sequence using two projection (proximal) operations per iteration (see, e.g., [3]). The algorithm in (9) performs instead one projection per iteration. Variations of the theme can be found in a number of references, including [2, 9, 27, 10]. Existing analyses however are not adequate to capture the behaviour of these methods in high-dimensions, including (9); as discussed in Sec. 1.1 for distributed implementations, they would

provide pessimistic predictions and break down under the scaling $d/N \to \infty$. As a by-product of our novel convergence analysis in the distributed setting, Theorem 1 below establishes, under RSC and RSM properties (Assumption 1), global convergence of the algorithm in (9) at linear (accelerated) rate to a fixed tolerance. This result is of independent interest, and complements existing accelerated schemes for $\ell_1$-least squares estimations (LASSO in Lagrangian form) [16, 31]. These are double-loop methods, combining accelerated proximal gradient with a homotopy continuation mechanism.

Leveraging Assumptions 1, 3, let us introduce the tolerance $\Delta^2$ and the initial optimality gap $V^0$:

$$\Delta^2 \triangleq \left( 2\mathcal{R} \left( \Pi_{\mathcal{M}^\perp}(\theta^\star) \right) + 2\mathcal{R}(\Delta^\star) + \Psi(\bar{\mathcal{M}}) \|\Delta^\star\| \right)^2, \quad V^0 \triangleq f(\theta^0) - f(\widehat{\theta}) + \frac{\mu}{8} \|v^0 - \widehat{\theta}\|^2, \quad (10)$$

where $\Delta^\star \triangleq \theta^\star - \widehat{\theta}$ is the satistical error, $\Pi_{\mathcal{M}^\perp}$ denotes the Euclidean projection onto the orthogonal complement of the subspace $\mathcal{M}$, and $\Psi(\bar{\mathcal{M}})$ is defined in Assumption 3. Convergence is stated next.

**Theorem 1.** *Given the ERM* (3), *suppose (i) $f$ satisfies the RSC/RSM conditions (Assumption 1); (ii) $\mathcal{R}$ is a decomposable regularizer with respect to chosen pair of subspaces $(\mathcal{M}, \bar{\mathcal{M}})$ (Assumption 3); and (iii) $(\mathcal{M}, \bar{\mathcal{M}})$ and RSC/RSM parameters $(\mu, \tau_\mu)$ and $(L, \tau_L)$ are such that $\Psi^2(\bar{\mathcal{M}}) \max\{\tau_L, \tau_\mu\} \leq C_0$, for some constant $C_0 > 0$. Let $\{\theta^t\}_{t\geq 1}$ be the sequence generated using Algorithm* (9), *with tuning $\alpha = 2L$ and $\beta = \sqrt{1/(8\kappa)}$. Then, for any solution $\widehat{\theta}$ of* (3) *for which $\mathcal{R}(\widehat{\theta}) = R$, we have*

$$\|\theta^t - \widehat{\theta}\|^2 \leq \frac{8}{\mu} \left( 1 - \sqrt{\frac{1}{16\kappa}} \right)^t V^0 + \mathcal{O} \left( \frac{\tau_\mu + \tau_L}{\mu} \Delta^2 \right). \quad (11)$$

This proves linear convergence at the accelerated rate $1 - \sqrt{1/(16\kappa)}$, up to a fixed tolerance. As it will be showed directly in the distributed setting (see Sec. 4), when considering specific statistical models under $\Psi^2(\bar{\mathcal{M}}) \max\{\tau_L, \tau_\mu\} = o(1)$–a condition necessary for statistical consistency [1, 29]–the residual error in (11) becomes *smaller than the statistical precision*, proving thus convergence to statistically optimal solutions. This improves the convergence rate of the PGA [1], with Algorithm (9) gaining acceleration while retaining the same statistical precision and computational complexity.

Finally, we remark that, while Theorem 1 has been stated for quadratic ERM, the same conclusions hold for nonquadratic losses, satisfying RSC/RSM conditions as in Assumption 1.

### 3.2 Distributed accelerated proximal gradient method for high-dimensional estimation

Equipped with Algorithm (9), we are ready to introduce our accelerated method for distributed high-dimensional estimations in the form (3), as given in Algorithm 1 below, and discussed next.

What prevents (9) from being implemented on mesh networks is the lack of knowledge of $\nabla f$ at the agents' sides. Agents' losses $f_i$'s are decoupled by introducing local estimates $\theta_i$, controlled (updated) by the intended agent, along with the associated sequences $z_i$ and $v_i$, which play the same role of $z$ and $v$ as in (9): first, they are locally updated as the related ones in (9), followed by the consensus step (16), enforcing agreement among the momentum sequences and local estimates. The update of the $v$-variables, as per (9) would require the knowledge of $\nabla f$ from each agent, an information that is not available. Hence, $\nabla f$ is replaced by a local estimate $y_i$ [see (13)]; the tracking of $\nabla f$ via each $y_i$ is employed by the dynamic consensus mechanism (a.k.a. gradient-tracking) [7, 30, 19] in (14). In fact, it is not difficult to to check that the average of the $y_i^t$'s tracks the average of the local gradients $\nabla f_i(z_i^t)$, i.e.,

$$\frac{1}{m} \sum_{i=1}^m y_i^t = \frac{1}{m} \sum_{i=1}^m \nabla f_i(z_i^t). \quad (12)$$

Assuming that consensus on the $z_i$'s and the $y_i$'s variables is asymptotically achieved, i.e., $\|z_i^t - z_j^t\| \to 0$ and $\|y_i^t - y_j^t\| \to 0$ as $t \to \infty$, for all $i, j = 1, \ldots m$ (a fact that will be proved), (12) would yield the desired gradient tracking property $\|\nabla f(z_i^t) - y_i^t\| \to 0$ as $t \to \infty$, for all $i = 1, \ldots, m$.

## 4 Statistical-Computational Guarantees

This section studies convergence of Algorithm 1. Our first result (Theorem 2) is of deterministic type: We establish conditions for the estimation error $(1/m) \sum_{i=1}^m \|\theta_i^t - \widehat{\theta}\|^2$ to shrink linearly

**Algorithm 1**

**Data:** Set $\theta_i^0 = v_i^0 = z^{-1} = 0$ and $y^{-1} = \nabla f_i(0)$ for all $i \in [m]$; $\alpha \geq 2L$ and $\beta \in (0, 1)$;

**Iterate** $t = 0, 1, \ldots$

**[S.1]:** Consensus and gradient-tracking: Each agent $i$ performs

$$z_i^t = \sum_{j \in \mathcal{N}_i} w_{ij} \left( \frac{1}{\beta + 1} \theta_j^t + \frac{\beta}{1 + \beta} v_j^t \right), \tag{13}$$

$$y_i^t = \sum_{j \in \mathcal{N}_i} w_{ij} \left( y_j^{t-1} + \nabla f_j(z_j^t) - \nabla f_j(z_j^{t-1}) \right); \tag{14}$$

**[S.2]:** Local Optimization and updates: Each agent $i$ performs

$$v_i^{t+1} = \Pi_{\Omega \cap \mathcal{R}(v) \leq R} \left( (1 - \beta) v_i^t + \beta z_i^t - \frac{\alpha}{\beta} y_i^t \right); \tag{15}$$

$$\theta_i^{t+1} = \beta \, v_i^{t+1} + (1 - \beta) \theta_i^t. \tag{16}$$

up to a tolerance. These conditions hold with high-probability for a variety of statistical models (Assumption 4), proving linear convergence (at accelerated rate) within minimax statistical rates.

**Theorem 2.** *Consider the ERM (3) over mesh networks, under the same assumptions (i)-(iii) as in Theorem 1; furthermore, (iv) suppose $f_i$ satisfies the local RSM condition (Assumption 2), for all $i \in [m]$. Let $\{(\theta_i^t)_{i \in [m]}\}_{t \geq 1}$ be the sequence generated by Algorithm 1, with tuning $\alpha = 2L$ and $\beta = \sqrt{1/(8\kappa)}$, and weight matrix $W$ satisfying Assumption 5, with $\rho$ such that*

$$\rho \leq C_1 \min \left\{ \sqrt{\kappa^{-1}}, \frac{L \, \kappa^{-1}}{m^{3/2} \left( \gamma_\ell + \Psi^2(\bar{\mathcal{M}})(\tau_\ell + \tau_L) \right)} \right\}, \tag{17}$$

*where $C_1 > 0$ is a universal constant (see Theorem 3 in Appendix A.4).*

*Then, for any solution $\widehat{\theta}$ of (3) for which $\mathcal{R}(\widehat{\theta}) = R$, we have: for some $C_2 > 0$,*

$$\frac{1}{m} \sum_{i=1}^m \|\theta_i^t - \widehat{\theta}\|^2 \leq \frac{8}{\mu} \left( 1 - \sqrt{\frac{1}{32\kappa}} \right)^t L^0 + \mathcal{O}\Big( \underbrace{\frac{\tau_\mu + \tau_L}{\mu} \Delta^2}_{\text{centralized precision}} + \underbrace{\frac{\tau_\ell \sqrt{m} \rho}{\mu} \Delta^2}_{\text{network error}} \Big). \tag{18}$$

*where $L^0 \in (0, \infty)$ captures the initial optimality gap and dependence on optimization and RSC/RSM parameters (its expression can be found in Theorem 3, Appendix A.4).*

This shows linear convergence at rate $1 - \sqrt{1/(32\kappa)}$ up to a tolerance. The first term of this tolerance matches that achieved in the centralized setting (see (11)) while the second term is an extra error due to the decentralization of the estimation process–it vanishes when $\rho = 0$ (fully connected network). In the next section, we will show that for the statistical models captured by Assumption (i), the overall tolerance can be made of a higher-order of the centralized statistical error. The condition on $\rho$ in (17) calls for a sufficiently connected network. When it is not satisfied by the given network and gossip matrix $W$, it can be enforced running multiple rounds of communications, affecting the total communication complexity by log-factor terms. More specifically, given $W$ with associated $\rho = \|W - J\|_2$ larger than what required in (17), one can build another matrix $\bar{W} = W^K$ and choose the positive integer number $K$ so that $\bar{\rho} \triangleq \|\bar{W} - J\|_2 = \rho^K$ satisfies (17); it is sufficient to set

$$K = \left\lceil \frac{\log \left( C_1 \kappa m^{2.5}(1 + C_2/L) \right)}{1 - \rho} \right\rceil.$$

The use of $\bar{W}$ instead of $W$ in Algorithm 1 corresponds to employing $K$ rounds of communications (consensus and tracking updates) per iteration $t$, each one using the weights $W$.

## 4.1 Statistical Guarantees

We apply now Theorem 2 to specific statistical models, establishing convergence and statistical guarantees with overwhelming probability. Corollary 1 studies the case of sparse linear regression,

under both strong and weak sparsity. Corollary 2 addresses the low-rank matrix recovery problem. Sub-Gaussian linear regression (Corollary 3) and matrix completion (Corollary 4) are discussed in the supporting material (Appendix A.3).

• **Sparse regression:** Suppose $\theta^\star$ in (1) is sparse, i.e., $\|\theta^\star\|_q \leq R_q$, with $q \in [0,1]$ ($q = 0$ corresponds to hard-sparsity, while $q > 0$ models weak sparsity). The regression model at agent $i$ is: $y_j = x_j^\top \theta^\star + w_j$, $j \in \mathcal{S}_i$, where $|\mathcal{S}_i| = n$, $w_j \sim \mathcal{N}(0, \sigma^2)$, and $x_j$ satisfies Assumption 4(i). Define

$$\chi_N \triangleq \frac{\max_i([\Sigma]_{ii})}{\lambda_{\min}(\Sigma)} R_q \left( \frac{\log d}{N} \right)^{1 - \frac{q}{2}}, \quad q \in [0,1] \text{ and } R_0 = s.$$

**Corollary 1.** *Consider the ERM 3 solving the linear regression problem above, with $\mathcal{R}(\cdot) = \| \cdot \|_1$, $\|\theta^\star\|_1 \leq R = \|\widehat{\theta}\|_1$, and $\Omega \equiv \mathbb{R}^d$; let $N = \Omega((\eta_\Sigma R_q)^{1/(1-q/2)} \log d)$. Let $\{(\theta_i^t)_{i=1}^m\}_{t \geq 0}$ be the sequence generated by Algorithm 1 with tuning as in Theorem 2, where (17) becomes*

$$\rho \leq C_1 \min \left\{ \sqrt{\kappa^{-1}}, \frac{\kappa^{-1}}{m^{2.5}(1 + C_2/L)} \right\}, \quad \text{for some } C_1, C_2 > 0. \tag{19}$$

*Then, with probability at least $1 - c_0 \exp(-c_1 N + \log(m))$, for some $c_0, c_1 > 0$, it holds*

$$\frac{1}{m} \sum_{i=1}^m \|\theta_i^t - \widehat{\theta}\|^2 \leq \frac{8}{\lambda_{\min}(\Sigma)} \left( 1 - \sqrt{\frac{1}{32\kappa}} \right)^t L^0 + \mathcal{O} \left( \chi_N \left( R_q \left( \frac{\log d}{N} \right)^{1 - \frac{q}{2}} + \|\Delta^\star\|^2 \right) \right), \tag{20}$$

*where $L^0 > 0$ depends on the the initial optimality gap and statistical parameters (see Appendix A.6)*

A few comments are in order. **(i)** Linear convergence at rate $1 - \mathcal{O}(\kappa^{-1/2})$ is certified, up to a tolerance that, under the sample-size $N = \Omega((\eta_\Sigma R_q)^{1/(1-q/2)} \log d)$, is of a higher order than the centralized statistical error $\|\Delta^\star\|^2$. In fact, the additional term $R_q(\frac{\log d}{N})^{1-q/2}$, due to the statistical nonidentifiability of linear regression over the $\ell_q$-ball and thus not improvable, is no larger than $\|\Delta^\star\|^2$ with high-probability. **(ii)** The aforementioned scaling on $N$ is unavoidable, as it matches that necessary for any (centralized) method to be statistically consistent over $\ell_q$-ball [23]. This proves that statistically optimal estimates are achievable over mesh networks even when the local sample size $n$ violates information theoretical lower bounds (as long as $N$ satisfies so). This is possible thanks to the network mixing local information at sufficiently fast rate, as required by (43). **(iii)** As anticipated, condition (43), when not implicitly satisfied, can be met running multiple rounds of communications/iteration; it can be shown that $\widetilde{\mathcal{O}}(\log m/(1 - \rho))$ are enough, resulting in an overall communication complexity as in (5) to reach an $\varepsilon$-neighborhood of a statistically optimal estimate.

• **Low-rank matrix recovery:** We consider the compressed sensing version of the matrix regression over meshed networks, as formulated next. Let $\Theta^\star$ in (1) satisfy $\|\Theta^\star\|_q \leq R_q$, with $q \in [0,1]$ and $R_q > 0$. In that the case $q = 0$ corresponds to matrices with rank at most $r$; we will have $R_0 = r$. The case $q > 0$ results in approximately low-rank of $\Theta^\star$. Consider observations $y_j = \langle X_j, \Theta^\star \rangle + w_j$, $j \in \mathcal{S}_i$, $|\mathcal{S}_i| = n$, where $X_j \in \mathbb{R}^{p \times p}$ is such that $\text{vec}(X_j)$ satisfies Assumption 4.(i), $w_j \sim \mathcal{N}(0, \sigma^2)$ are i.i.d. and independent of $X_j$. Define

$$\chi_N \triangleq \frac{\max_i([\Sigma]_{ii})}{\lambda_{\min}} R_q \left( \frac{p}{N} \right)^{1 - \frac{q}{2}}, \quad q \in [0,1] \text{ and } R_0 = r.$$

**Corollary 2.** *Consider the ERM (3) solving the low-rank matrix regression problem above with $\mathcal{R}(\Theta) = \|\Theta\|_1$, $\|\Theta^\star\|_1 \leq R = \|\widehat{\theta}\|_1$, and $\Omega \equiv \mathbb{R}^{p \times p}$; let $N = \Omega\left(p(\eta_\Sigma R_q)^{1/(1-q/2)}\right)$. Let $\{(\theta_i^t)_{i=1}^m\}_{t \geq 0}$ be the sequence generated by Algorithm 1 with tuning as in Theorem 2, including (17) for some $C_1, C_2 > 0$. Then, with probability at least $1 - c_0 \exp(-c_1 N + \log(m))$, for some $c_0, c_1 > 0$, it holds:*

$$\frac{1}{m} \sum_{i=1}^m \|\theta_i^t - \widehat{\theta}\|^2 \leq \frac{8}{\lambda_{\min}(\Sigma)} \left( 1 - \sqrt{\frac{1}{32\kappa}} \right)^t L^0 + \mathcal{O} \left( \chi_N \left( R_q \left( \frac{p}{N} \right)^{1 - \frac{q}{2}} + \|\Delta^\star\|^2 \right) \right),$$

*where $L^0 > 0$ depends on the the initial optimality gap and statistical parameters (see Appendix A.6).*

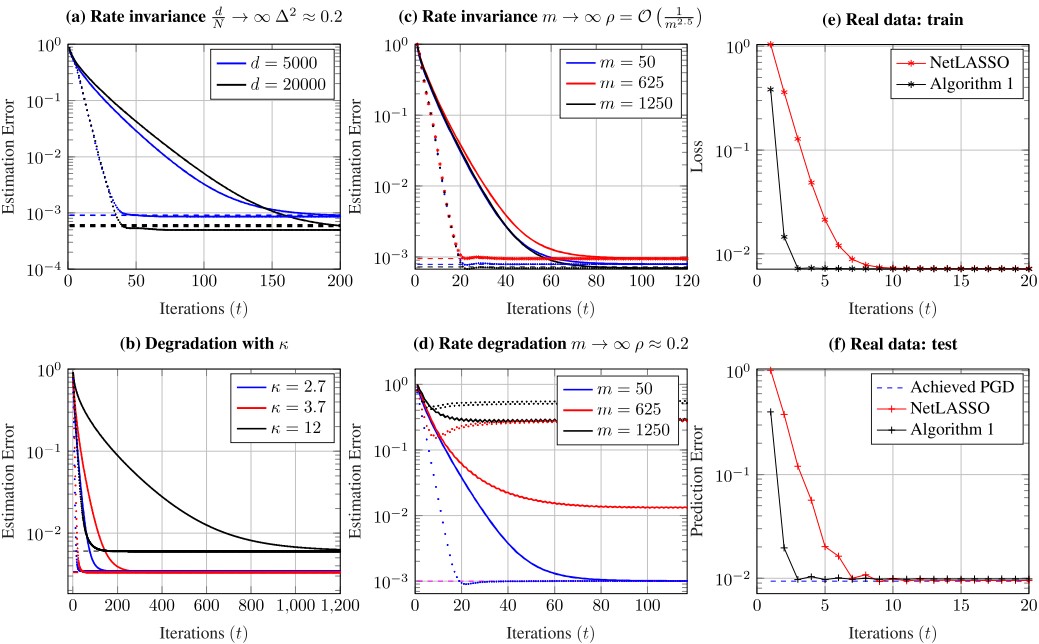

Figure 2: Algorithm 1 (dotted-lines) vs. NetLASSO (solid-lines). **(a)** Linear convergence vs iterations up to statistical precision (dashed) $\Delta^2$, for $s, d, N$ growing, such that $\Delta^2 \approx 0.2$. **(b)** Estimation error vs iterations for different values of $\kappa$. **(c)** Estimation error vs iterations for growing $m$ such that $\rho = \mathcal{O}(m^{-2.5})$. **(d)** Same as (c) but with $\rho = \approx 0.2$. **(e)-(f):** `eyedata` data set, (e) training error vs. iterations, (f) testing error vs. iterations. Although quantitative aspects of the sample rates are different, Corollary 2 is analogous to Corollary 1, which we refer to for specific comments. As therein, also here the final tolerance is of a higher-order of the statistical error of the model.

Overall the above statistical guarantees (along with those discussed in the supporting material for other estimation problems, see Appendix A.3) complement [1], matching those therein but over meshed networks while improving convergences rate of the PGA gaining acceleration.

## 5 Numerical Results

Numerical results are presented in this section validating our theoretical findings on synthetic and real data. Simulations are performed on a computer with an Intel i7-8650U CPU @ 1.9 GHz using 16 GB RAM.

**Synthetic data:** We simulate the sparse linear regression model as in Corollary 1, with $q = 0$. We set $R_0 = s$, $\sigma^2 = 0.25$ and $R = \|\theta^\star\|^2$. The covariates $x_i \in \mathbb{R}^d$ are generated as in [1] $x_{i,j} = z_{i,j} + \omega x_{i,j-1}$, with $x_{i,1} = z_{i,1}$, where each $z_{i,j} \sim \mathcal{N}(0,1)$ and i.i.d. This implies that $\mu \geq \mu' \triangleq (1+\omega)^{-2}$ and $L \leq L' \triangleq 2((1+\omega)^2(1+\omega))^{-1}$. We will denote $\Delta^2 \triangleq (s \log d)/N$, which is of the order of the statistical precision of the model. The network is generated using an Erdős-Rényi graph, with $m$ agents and link activation probability $p$. The specific values of each of the parameters is given in each specific experiment. All curves are averaged over 10 independent Montecarlo simulations. We now proceed to verify the different aspects of Corollary 1.

**1) Linear convergence to optimal estimates:** Fig 2.(a) plots the normalized estimation error $\sum_i \|\theta_i^t - \theta^\star\|^2/\|\theta^\star\|^2$ versus iterations, generated by Algorithm 1 (dashed-lines) and NetLASSO [26] (solid lines), solving the regression problem, with $\omega = 0.5$ (yielding $\mu = 0.44$ and $L = 5.33$); different curves refer to different values of $s, d, N$, with $d$ growing faster than $N$, such that $\Delta^2 \approx 0.2$, and $m = 5$. In both algorithms we set the stepsize $\alpha = 2L'$; the momentum parameter $\beta$ is set to $\beta = (8\mu'/L')^{-1/2}$. Condition (43) is enforced via $T = 41$ rounds of communications/iteration for both algorithms. In **Fig. 2.(b)** we plot the same quantities now for different condition numbers $\kappa$, obtained choosing $\omega \in \{0.1, 0.2, 0.5\}$. The following comments are in order: **(i)** As predicted by (20), Algorithm 1 convergences at linear rate to statistically optimal estimates; although it has the same computational cost per iteration of the nonaccelerated NetLASSO, it is much faster, especially when $\Sigma$ is ill-conditioned (large $\kappa$), which proves the advantages of acceleration also in high-dimensions.

**(ii)** As $d$ and $N$ increase, as long as $\Delta^2$ is constant, the rate and reached statistical accuracy remain roughly invariant. (Fig. 2.(a)), which is a resirable property.

**2) On the scaling of $\rho$ with $m$ as in (43): Fig. 2.(c)-(d)** plots the same quantities as in Fig. 2(a), now with fixed $d = 5000$, $s = \lceil \sqrt{d} \rceil$, $N = n \cdot m = 2500$, and varying $m \in \{50, 625, 1250\}$. The random graph is generated so that in plot (c) $\rho \approx \mathcal{O}(m^{-2.5})$ (this is achieved setting $p = 0.8$) while in plot (d) $\rho \approx 0.2$ for all $m$ (achieved setting $p \in \{0.7, 0.4, 0.22\}$ for the associated values of $m$). The number of communications/iteration is set to 7 for both algorithms. The two plots show that $\rho$ should decrease with $m$ if one wants to achieve centralized statistical consistency; $\rho \approx \mathcal{O}(m^{-2.5})$, as predicted by (43), seems to be sufficient. On the other hand, a $\rho$ constant with $m$ breaks down the performance of both distributed algorithms.

**(B) Real data:** We test here NetLASSO and Algorithm 1 on the data set `eyedata` in the `NormalBetaPrime` package [4]; $d = 200$ and $N = 120$. We generated $\bar{W}$ for $m = 10$ and $p = 0.9$. To achieve the required connectivity we set $W = \bar{W}^7$. The data set is split between test- and training-data, corresponding to $N_{\text{test}} = 40$ and $N_{\text{train}} = 80$, respectively. Both data sets are equally distributed across $m = 10$ agents. In Fig. 2.(e) we display quantitites relevant to the training phase. Specifically, we plot the objective function value along the iterates generated by NetLASSO and Algorithm 1. We observe that both schemes achieve a similar final objective function value and converge linearly, with Algorithm 1 at much faster rate. Fig. 2.(f) shows the performance on the test data. We implemented the following procedure. Denote by $y$ the output of the test-set and by $X$ the test covariates. We build predictors $y_i^t = X \theta_i^t$, $i \in [m]$, where $\theta_i^t$ is the estimate of $\theta^\star$ from agent $i$ at time $t$, generated by the two algorithms when running on the training set. The figure plots $(N_{\text{test}} m)^{-1} \sum_{i=1}^m \|y - y_i^t\|_2^2$ versus iterations, for NetLASSO and Algorithm 1. The dashed line corresponds to $(N_{\text{test}} m)^{-1} \|y - \hat{y}\|^2$, where $\hat{y} = X\widehat{\theta}$ and $\widehat{\theta}$ denotes the estimator obtained via the PGA. The performance on test-data are consistent with what observed already on training-data.

# 6 Conclusions

We proposed a decentralized accelerated algorithm to solve quadratic high dimensional estimation problems over mesh networks whose empirical losses are neither strongly convex nor Lipschitz smooth globally. To employ acceleration in this setting, the design hinges on careful considerations regarding the directions traversed by the schemes, enforcing sparsity of the iterate and momentum sequences. Global convergence to statistically optimal solutions is proved at liner rate, proportional to $\sqrt{\kappa}$, with a communication cost per iteration at most $\tilde{\mathcal{O}}\left(\log m / (1 - \rho)\right)$. It is unclear whether this communication cost is improvable, for instance, whether the log-dependence on the number of agents and the transmission of all $d$ gradient/iterate components can be relaxed. This is left to future investigations.

## Acknowledgments and Disclosure of Funding

Funding in direct support of this work: ONR Grant N. N00014-21-1-2673.

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
