# Appendix

## A  Appendix

This appendix contains supplementary theoretical results as well as the proofs of the main results presented in the main paper. We organize this content as follows.

Sec. A.1 contains the list of the notation used in the paper and the rest of this appendix.

Sec. A.2 contains additional numerical simulations.

Sec. A.3 complements results in Sec. 4.1, establishing guarantees of Algorithm 1 applied to sub-Gaussian linear regression (Corollary 4) and convex matrix completion (Corollary 3).

Sec. A.4 Contains a more general version of Theorem 2 along with its proof..

Sec. A.6 Contains proofs of Corollaries 1-4

Sec. A.7 Contains some intermediate technical results instrumental for the proofs presented in Section A.4.

### A.1  Notation

Problem size:

| Symbols | Location | Description |
|---|---|---|
| $d$ | (1) | Problem dimension |
| $n$ | (2) | Number of local samples |
| $m$ | (3) | Number of agents |
| $N = n \cdot m$ | Section 1 | Total number of samples |

Relevant functions:

| Symbols | Location | Description |
|---|---|---|
| $f_i$ | (2) | Local empirical loss |
| $f$ | (3) | Global empirical loss |
| $\boldsymbol{f}$ | (21) | Stacked empirical risk |
| $\boldsymbol{F}$ | (22) | Stacked global empirical risk |

Population curvature parameters:

| Symbols | Location | Description |
|---|---|---|
| $\kappa$ | Section 1 | $\frac{\lambda_{\max}(\Sigma)}{\lambda_{\min}(\Sigma)}$ (See Section A.4) |

Network quantities:

| Symbols | Location | Description |
|---|---|---|
| $\rho$ | Assumption 5 | Network connectivity |
| $W$ | Assumption 5 | Gossip matrix |
| $\mathbf{W}$ | Section A.4 | $W \otimes I_d$ |
| $J$ | Section 5 | $J = \frac{1}{m}\mathbf{1}_m\mathbf{1}_m^\top$ |
| $\mathbf{J}$ | Section A.4 | $\mathbf{J} = J \otimes I_d$ |

Structure promotion:

| Symbols | Location | Description |
|---|---|---|
| $\mathcal{R}$ | (3) | Norm constraint |
| $R$ | (3) | Constraint radius |
| $\mathcal{M}$ | Assumption 3 | Model subspace |
| $\bar{\mathcal{M}}^\perp$ | Assumption 3 | $\mathcal{R} - \ell_2$ Lipschitz constant $\bar{\mathcal{M}}$ |
| $\mathcal{R}^*$ | Section A.4 | Dual norm to $\mathcal{R}$ |

Tolerances:

| Symbols | Location | Description |
|---|---|---|
| $\gamma_\ell, \tau_\ell$ | Assumption 2 | Local RSM constant and tolerance |
| $\mu, \tau_\mu$ | Assumption 1 | Global RSC constant and tolerance |
| $L, \tau_L$ | Assumption 1 | Global RSM constant and tolerance |

Other symbols:

| Symbols | Location | Description |
|---|---|---|
| $\mathcal{S}_i$ | (2) | Indexes of samples corresponding to agent $i$ |
| $\alpha$ | Algorithm 1 | Step-size |
| $\beta$ | Algorithm 1 | Momentum term |

We boldface stacked quantities of any type, vector and matrices. For vectors, the stacked is obtained grouping agents' variables into a single vector; for instance, we write $\boldsymbol{\theta} \triangleq [\theta_1^\top, \ldots, \theta_m^\top]^\top$, where $\theta_i$ is the local variable owned by agent $i$. For gossip matrices $A \in \mathbb{R}^{m \times m}$ we define their "augmented" instance as

$$\mathbf{A} \triangleq A \otimes I_d,$$

where $\otimes$ denotes the Kronecker product.

Following the same rationale, we define the stacked local empirical risk as

$$\boldsymbol{f}(\boldsymbol{\theta}) \triangleq \frac{1}{m} \sum_{i=1}^{m} f_i(\theta_i), \tag{21}$$

while the stacked average loss is defined as

$$\boldsymbol{F}(\boldsymbol{\theta}) \triangleq \frac{1}{m} \sum_{i=1}^{m} f(\theta_i). \tag{22}$$

Other useful notation is the following: $1_m \in \mathbb{R}^m$ is the vector of all 1's. We define $J \triangleq \frac{1}{m} 1_m 1_m^\top$; it is the projection onto the consensus space. Given a matrix $\Sigma$, we denote by $\lambda_i(\Sigma)$ and $\sigma_i(\Sigma)$ its $i^{\text{th}}$ largest eigenvalue and singular value, respectively–with $\lambda_{\max}(\Sigma)$ and $\lambda_{\min}(\Sigma)$ (resp. $\sigma_{\max}(\Sigma)$ and $\sigma_{\min}(\Sigma)$) being the largest and smallest eigenvalues (resp. singular values), respectively. We denote by $\mathcal{R}^*$ the dual norm of the norm $\mathcal{R}$. For any $\Theta \in \mathbb{R}^{p \times p}$, $\|\Theta\|_1 = \sum_{j=1}^{p} \sigma_j(\Theta)$ is the nuclear norm; $\|\Theta\|_\infty = \max_{i,j \in [p]} |\Theta_{i,j}|$; $\|\|\Theta\|\|_\infty = \max_{i \in [p]} \sum_{j \in [p]} |\Theta_{i,j}|$; $\|\|\Theta\|\|_2$ is the spectral norm; and $\|\Theta\|_F$ is the Frobenius norm. Finally, denote by $\|\mathbf{x}\|_{\mathbf{A}}^2$ the semi-norm induced by the p.s.d. matrix $\mathbf{A}$, i.e. $\|\mathbf{x}\|_{\mathbf{A}}^2 \triangleq \langle \mathbf{x}, \mathbf{A}\mathbf{x} \rangle$.

### A.2 Additional Experiments

This section complements the results presented in Sec. 5. In particular we show the invariance of the performance of Algorithm 1 under the asymptotic scaling "$d/N$ growing", as long as $\frac{s \log d}{N}$ remains constant and provide details on the results in Fig 1.

Consider the sparse linear regression model as described in Sec. 5, with $q = 0$, $\sigma^2 = 0.25$ and $R = \|\theta^\star\|^2$. The covariates $x_{i,j} \sim \mathcal{N}(0,1)$ are i.i.d. leading to $\mu = L = 1$. All the curves are averaged over 10 independent Montecarlo simulations. The network is generated using an Erdős-Rényi graph, with $m = 5$ and $p = 0.5$. To achieve a connectivity $\rho \approx 0.02$ we run 14 rounds of gradient tracking and consensus per iteration. Figure 3 plots the normalized estimation error $\frac{\sum_{i=1}^{m} \|\theta_i^t - \theta^\star\|^2}{m\|\theta^\star\|^2}$ versus iterations, generated by Algorithm 1 and NetLASSO (solid lines) solving the regression problem with $d = \{10^4, 10^5, 10^6\}$, $n = \{80, 100, 120\}$, $\alpha = 1/2$ and $\beta = \sqrt{1/8}$. We observe that as predicted by the theory, the achieved statistical precision and convergence rate remain of the same order, under the scaling $\frac{d}{N} \uparrow$. Furthermore, Figure 3 shows a gain using acceleration in high dimension even when $\kappa = 1$.

**Real data:** We test the NetLASSO and Algorithm 1 on the data set `Housing`[1], with $d = 13$, and total sample size $N_{\text{test+train}} = 500$. The network is generated using an Erdős-Rényi graph with $m = 5$

---

[1]The data set can be found `https://www.cs.ubc.ca/ schmidtm/Software/lasso.html`

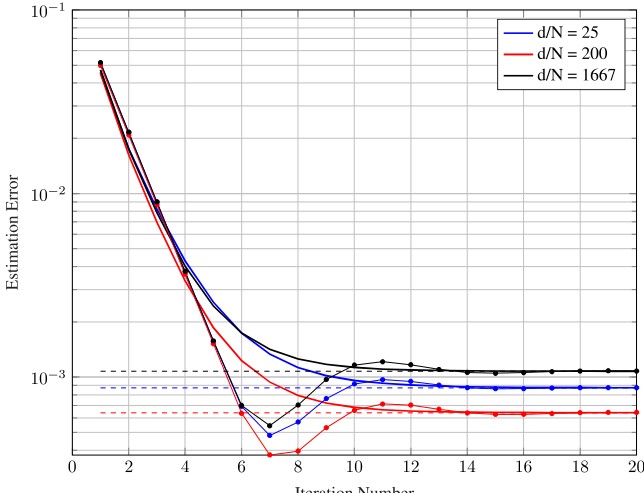

Figure 3: Estimation error versus interactions for different values of $\frac{d}{N}$; $d = 10^4$, $d^5$, and $d^6$; $n = 80$, 100 and 120, with $m = 5$. The network has connectivity $\rho = 0.023$; $\mu = L = 1$, the stepsize is then set to $1/2$, and $\beta = \sqrt{1/8}$. Solid line curves refer to NetLASSO while dotted-line curves represent the accelerated variant. Observe that even when the population has $\kappa = 1$ acceleration yields a gain (as predicted by the theory). Dotted-line curves indicate statistical precision achieved using centralized PGD.

and $p = 0.5$; 14 communication rounds are performed to achieve a connectivity of $\rho \approx 0.02$. The data set is split between training and test samples, $N_{\text{train}} = 10$ and $N_{\text{test}} = 490$, respectively. We denote by $X_i^{\text{train}}, y_i^{\text{train}}$ (resp. $X_i^{\text{test}}, y_i^{\text{test}}$) the training (resp. testing) sample pairs assigned to agent $i$ . Observe that we choose $N_{\text{train}}$ small to illustrate the performance when the objective function is not strongly convex. The left plot in Figure 4 reports the performance in terms of

$$\frac{\sum_{i=1}^{m} \left\| X_i^{\text{train}} \theta_i^t - y_i^{\text{train}} \right\|^2}{\sum_{i=1}^{m} \left\| X_i^{\text{train}} \bar{\boldsymbol{\theta}} - y_i^{\text{train}} \right\|^2}$$

where $\bar{\boldsymbol{\theta}}$ is the solution obtained via PGD (centralized), and $\theta_i^t$ denote the iterates generated by either NetLASSO (black-line curves) or Algorithm 1 (red-line curves). The right plot in Figure 4 shows the performance of the two algorithms in terms of test/prediction error, defined as

$$\frac{\sum_{i=1}^{m} \left\| X_i^{\text{test}} \theta_i^t - y_i^{\text{test}} \right\|^2}{\sum_{i=1}^{m} \left\| X_i^{\text{train}} \bar{\boldsymbol{\theta}} - y_i^{\text{train}} \right\|^2},$$

where $\theta_i^t$ are obtained using the NetLASSO and Algorithm 1 (black-line and red-line curves, respectively) using the train data set, where $\bar{\theta}$ is obtained using the centralized PGD on the train data set. Observe that both test and train errors are well behaved while the test error ends up being slightly larger than the train error.

The experiment in Fig 1 has set-up identical to those in Section 5 with $d = 20000$, $n = 40$, $m = 5$, $\rho = 0.023$ and $\sigma = 0.25$. $\beta = \sqrt{\frac{\mu}{8L}}$ for both DPAG and our proposed scheme, while the step-size is set to $(2L)^{-1}$ for all schemes involved.

### A.3 Additional Statistical Models and guarantees

This section contains statistical-computation guarantees of Algorithm 1 applied to (i) the sparse linear regression, with covariantes $x_i$ drawn from a sub-Gaussian distribution; and (ii) the convex matrix completion problem.

• **Sparse sub-Gaussian regression:** consider the sparse regression model described in Section 4 where $x_i$ are i.i.d. and now $(\tau^2, \Sigma)-$sub-Gaussian, i.e., fulfill Assumption 4.(ii). Denote by $X \triangleq$

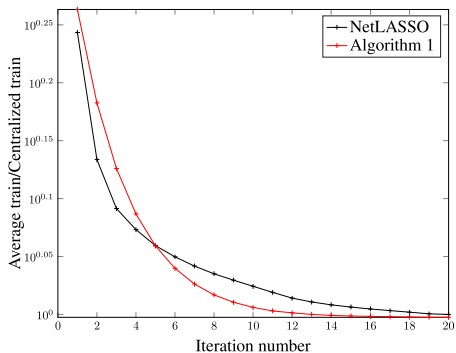 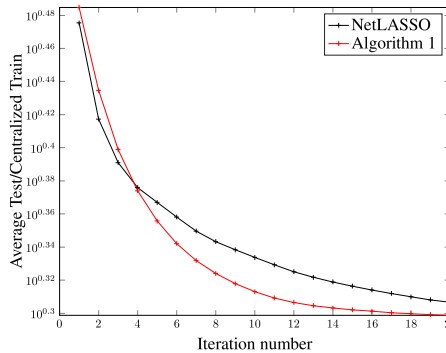

Figure 4: `Housing` data set. Left panel: normalized training error v.s. iterations; Right panel: normalized testing error v.s. iterations.

$[x_1^\top; \ldots; x_N^\top]$ the matrix containing the samples $x_i$ row-wise, and assume that each column of $X$ has $\ell_2$-norm equal to $C > 0$. This assumption is standard when dealing with sub-Gaussian measurement matrices [29]. Finally, define

$$\chi_N \triangleq \frac{\max\left\{\frac{\tau^4}{\lambda_{\min}(\Sigma)}, \lambda_{\min}(\Sigma)\right\}}{\lambda_{\min}(\Sigma)} R_q \left(\frac{\log d}{N}\right)^{1-\frac{q}{2}}, \quad q \in [0,1] \text{ and } R_0 = s.$$

**Corollary 3.** *Consider the ERM* (3) *solving the linear regression problem described above, with* $\mathcal{R}(\cdot) = \|\cdot\|_1, \|\theta^\star\|_1 \le R = \|\widehat{\theta}\|_1$ *and* $\Omega \equiv \mathbb{R}^d$; *and let*

$$N = \Omega\left((R_q \max\left\{\frac{\tau^4}{\lambda_{\min}(\Sigma)}, \lambda_{\min}(\Sigma)\right\})^{1/(1-q/2)} \log d\right).$$

*Let* $\{(\theta_i^t)_{i=1}^m\}_{t\ge 0}$ *be the sequence generated by Algorithm 1 with tuning as in Theorem 2, where* (17) *becomes*

$$\rho \le C_1 \min\left\{\sqrt{\kappa^{-1}}, \frac{\kappa^{-1}}{m^{2.5}(1 + C_2/L)}\right\}, \quad \text{for some } C_1, C_2 > 0.$$

*Then, with probability at least* $1 - c_0 \exp\left(-c_1 N \min\{\mu^2/\tau^4, 1\} + \log(m)\right)$, *for some* $c_0, c_1, > 0$, *it holds*

$$\frac{1}{m}\sum_{i=1}^m \|\theta_i^t - \widehat{\theta}\|^2 \le \frac{8}{\lambda_{\min}(\Sigma)}\left(1 - \sqrt{\frac{1}{32\kappa}}\right)^t L^0 + \mathcal{O}\left(\chi_N\left(R_q\left(\frac{\log d}{N}\right)^{1-\frac{q}{2}} + \|\Delta^\star\|^2\right)\right).$$

*Proof.* See Appendix A.6.3. $\square$

● **Matrix Completion:** Let $y_j$ be a noisy measurement of $[\Theta^\star]_{a(j),b(j)}$, which corresponds to the $a(j), b(j)$ element of the matrix $\Theta^\star \in \mathbb{R}^{p\times p}$, which is selected uniformly at random from its $d = p^2$ elements. $\Theta^\star$ is assumed to be near low-rank, in the sense that $\|\Theta^\star\|_q \le R_q$ with $q \in (0,1]$ where $\|\Theta^\star\|_q \triangleq \sum_{i=1}^p \sigma_i(\Theta^\star)^q$. Let $X_j \in \mathbb{R}^{p\times p}$, with $[X_j]_{a(j),b(j)} = 1$ and remaining entries equal to zero. Then, $y_j = \langle X_j, \Theta^\star\rangle + (\sigma/d)w_j, j \in \mathcal{S}_i, i \in [m]$, and $w_j$ are sub-exponential with parameter 1 and independent from $X_i$. Our goal is to compute an estimate of $\Theta^\star$. To achieve this, we cast the problem into an ERM of the form (3) where we choose $\Omega = \{\Theta \in \mathbb{R}^{p\times p} : \|\Theta\|_\infty \le \omega/p\}$ with $\omega \ge 1$. Observe that by vectorizing the quantities involved in the inner product, Assumption 4(iii) is fulfilled.

**Corollary 4.** *Consider the ERM 3 solving the matrix completion described above, with* $\mathcal{R}(\cdot) = \|\cdot\|_1$, $\|\Theta^\star\|_1 \le R = \|\widehat{\Theta}\|_1$ *and* $\Omega = \{\Theta : \|\Theta\|_\infty \le \frac{\omega}{p}\}$, *with* $\omega \ge 1$; *and set* $N = \Omega\left(p \log p\right)$. *Let* $\{(\Theta_i^t)_{i=1}^m\}_{t\ge 0}$ *be the sequence generated by Algorithm 1 with tuning as in Theorem 2, where* (17) *becomes*

$$\rho \le \frac{C_1}{m^5}, \quad \text{for some } C_1 > 0.$$

*Then, with probability at least* $1 - c_0(m+1)\exp\left(-c_1 p \log p\right)$ *for some* $c_0, c_1 > 0$ *it holds*

$$\frac{1}{m}\sum_{i=1}^m \|\theta_i^t - \widehat{\theta}\|^2 \leq 8\left(1 - \frac{1}{4\sqrt{2}}\right)^t L^0 + \mathcal{O}\left(R_q \omega^{2-q}\left(\frac{p\log p}{N}\right)^{1-\frac{q}{2}} + \|\Delta^\star\|^2\right).$$

*Proof.* See Appendix A.6.4. □

## A.4  Proof of Theorem 2

Using the notation introduced in Appendix A.1, we start rewriting Algorithm 1 in the "augmented" vector-matrix form:

$$\mathbf{w}^t = \frac{1}{\beta+1}\boldsymbol{\theta}^t + \frac{\beta}{1+\beta}\mathbf{v}^t \tag{23a}$$

$$\mathbf{z}^t = \mathbf{W}\mathbf{w}^t \tag{23b}$$

$$\mathbf{y}^t = \mathbf{W}\left(\mathbf{y}^{t-1} + \left(\nabla \boldsymbol{f}(\mathbf{z}^t) - \nabla \boldsymbol{f}(\mathbf{z}^{t-1})\right)\right) \tag{23c}$$

$$\mathbf{v}^{t+1} = \operatorname*{argmin}_{v_i \in \Omega : \mathcal{R}(v_i) \leq R, \forall i}\left\{m\langle \mathbf{y}^t, \mathbf{v}\rangle + \frac{\beta\alpha}{2}\|\mathbf{v} - (1-\beta)\mathbf{v}^t - \beta\mathbf{z}^t\|^2\right\} \tag{23d}$$

$$\boldsymbol{\theta}^{t+1} = \beta\mathbf{v}^{t+1} + (1-\beta)\boldsymbol{\theta}^t, \tag{23e}$$

where $\mathbf{z}^{-1} = \mathbf{W}\mathbf{w}^{-1}$, and $\mathbf{w}^{-1}$ is any arbitrary vector, with each of the $d$-component being feasible for (3). Notice that, with a slight abuse of notation, we still used the (stacked) vector $\mathbf{y}$ to denote the tracking variables as in (23c), although in (23d) they are $m$ times those defined in (14), due to the $1/m$ factor in the definition of $\boldsymbol{f}$ (see (21)). This explains the coefficient $m$ in front of $\langle \mathbf{y}^t, \mathbf{v}\rangle$ in (23d).

We will study convergence leveraging the following Lyapunov-like function along the iterates of the algorithm:

$$V^t \triangleq \boldsymbol{F}(\mathbf{W}\boldsymbol{\theta}^t) - \boldsymbol{F}(\mathbf{W}\widehat{\boldsymbol{\theta}}) + \frac{\beta^2\alpha}{2m}\|\boldsymbol{\theta}^t\|_{\mathbf{I}-\mathbf{W}^2}^2 + \frac{\beta^2\alpha}{2m}\|\mathbf{v}^t - \widehat{\boldsymbol{\theta}}\|^2. \tag{24}$$

For convenience, define

$$\boldsymbol{g}(\mathbf{w}) \triangleq \boldsymbol{F}(\mathbf{W}\mathbf{w}) + \frac{\beta^2\alpha}{2m}\|\mathbf{w}\|_{\mathbf{I}-\mathbf{W}^2}^2.$$

It is not difficult to check that, under Assumption 1, $\mathbf{g}$ inherits the RSC and RSM properties of $f$, in the following sense:

$$\frac{\mu}{4m}\|\mathbf{x} - \mathbf{y}\|_{\mathbf{W}^2}^2 + \frac{\beta^2\alpha}{2m}\|\mathbf{x} - \mathbf{y}\|_{\mathbf{I}-\mathbf{W}^2}^2 - \varepsilon_\ell(\mathbf{x}, \mathbf{y}) \leq \boldsymbol{g}(\mathbf{x}) - \boldsymbol{g}(\mathbf{y}) - \langle \nabla \boldsymbol{g}(\mathbf{y}), \mathbf{x} - \mathbf{y}\rangle, \tag{25a}$$

$$\boldsymbol{g}(\mathbf{x}) - \boldsymbol{g}(\mathbf{y}) - \langle \nabla \boldsymbol{g}(\mathbf{y}), \mathbf{x} - \mathbf{y}\rangle \leq \frac{L}{m}\|\mathbf{x} - \mathbf{y}\|_{\mathbf{W}^2}^2 + \frac{\beta^2\alpha}{2m}\|\mathbf{x} - \mathbf{y}\|_{\mathbf{I}-\mathbf{W}^2}^2 + \varepsilon_u(\mathbf{x}, \mathbf{y}), \tag{25b}$$

for all $\mathbf{x}, \mathbf{y} \in \Omega^m$, where

$$\varepsilon_\ell(\mathbf{x}, \mathbf{y}) \triangleq \frac{\tau_\mu}{2m}\sum_{i=1}^m \mathcal{R}\left(\sum_{j=1}^m w_{i,j}(x_j - y_j)\right) \text{ and } \varepsilon_u(\mathbf{x}, \mathbf{y}) \triangleq \frac{\tau_L}{2m}\sum_{i=1}^m \mathcal{R}^2\left(\sum_{j=1}^m w_{i,j}(x_j - y_j)\right). \tag{26}$$

Using the above properties, our first step is to establish descent on the Lyapunov function up to some error; formally, we have the following.

**Proposition 1.** *For any* $\alpha \geq 2L$, *the following holds:*

$$V^{t+1} \leq (1-\beta)V^t - \frac{\beta}{2m}\left(\frac{\mu}{2} - \beta^2\alpha\right)\|\mathbf{z}^t - \widehat{\boldsymbol{\theta}}\|^2$$
$$+ \beta\varepsilon_\ell(\widehat{\boldsymbol{\theta}}, \mathbf{w}^t) + \varepsilon_u(\boldsymbol{\theta}^{t+1}, \mathbf{w}^t) + \beta\langle \boldsymbol{\varepsilon}^t, \widehat{\boldsymbol{\theta}} - \mathbf{v}^{t+1}\rangle, \tag{27}$$

*for all* $t = 0, 1, \ldots$, *where*

$$\boldsymbol{\varepsilon}^t \triangleq \mathbf{y}^t + \frac{\beta^2\alpha}{m}(\mathbf{W} - \mathbf{I})\mathbf{z}^t - \mathbf{W}\nabla \boldsymbol{F}(\mathbf{W}\mathbf{w}^t). \tag{28}$$

*Proof.* See Appendix A.7.1. □

The rest of the proof consists of controlling the error terms

$$\beta\varepsilon_\ell(\widehat{\boldsymbol{\theta}}, \mathbf{w}^t) + \varepsilon_u(\boldsymbol{\theta}^{t+1}, \mathbf{w}^t) + \beta\langle\boldsymbol{\varepsilon}^t, \widehat{\boldsymbol{\theta}} - \mathbf{v}^{t+1}\rangle.$$

The errors $\varepsilon_\ell(\widehat{\boldsymbol{\theta}}, \mathbf{w}^t)$ and $\varepsilon_u(\boldsymbol{\theta}^{t+1}, \mathbf{w}^t)$ can be controlled using the decomposability of the regularizer–Lemma 2 below provides the desired bound–whereas $\langle\boldsymbol{\varepsilon}^t, \widehat{\boldsymbol{\theta}} - \mathbf{v}^{t+1}\rangle$ requires a suitable analysis of the tracking variables $\mathbf{y}^t$–Proposition 2 below summarizes the result.

**Lemma 2.** *Let*

$$\nu^2 \triangleq 2\left(2\mathcal{R}\left(\Pi_{\mathcal{M}^\perp}(\theta^\star)\right) + 2\mathcal{R}(\theta^\star - \widehat{\theta}) + \Psi(\bar{\mathcal{M}})\|\widehat{\theta} - \theta^\star\|\right)^2. \tag{29}$$

*The following bounds hold for $\varepsilon_\ell(\widehat{\boldsymbol{\theta}}, \mathbf{w}^t)$ and $\varepsilon_u(\boldsymbol{\theta}^{t+1}, \mathbf{w}^t)$:*

$$\varepsilon_\ell(\widehat{\boldsymbol{\theta}}, \mathbf{w}^t) \leq \frac{1}{2m}\tau_\mu 8\Psi^2(\bar{\mathcal{M}})\|\mathbf{z}^t - \widehat{\boldsymbol{\theta}}\|^2 + \frac{\tau_\mu}{2}\nu^2, \tag{30}$$

$$\varepsilon_u(\boldsymbol{\theta}^{t+1}, \mathbf{w}^t) \leq \frac{1}{2m}24\tau_L\Psi^2(\bar{\mathcal{M}})\beta^2\left(\|\mathbf{v}^{t+1} - \widehat{\boldsymbol{\theta}}\|^2 + \beta^2\|\mathbf{w}^t - \widehat{\boldsymbol{\theta}}\|^2_{\mathbf{W}^2} + (1-\beta)^2\|\mathbf{v}^t - \widehat{\boldsymbol{\theta}}\|^2\right)$$

$$+ \frac{\tau_L}{2}9\beta^2\nu^2. \tag{31}$$

*Proof.* See Appendix A.7.2. □

**Proposition 2.** *The following bound holds for $\langle\boldsymbol{\varepsilon}^t, \widehat{\boldsymbol{\theta}} - \mathbf{v}\rangle$:*

$$\langle\boldsymbol{\varepsilon}^t, \widehat{\boldsymbol{\theta}} - \mathbf{v}\rangle \leq \frac{A(\rho)}{2m}\|\mathbf{v} - \widehat{\boldsymbol{\theta}}\|^2 + \frac{A(\rho)}{2m}\|\mathbf{z}^t - \widehat{\boldsymbol{\theta}}\|^2 + \frac{B(\rho)}{2m}\|\boldsymbol{\theta}^t\|^2_{\mathbf{I}-\mathbf{W}^2} + \frac{B(\rho)}{2m}\beta\|\mathbf{v}^t - \widehat{\boldsymbol{\theta}}\|^2$$

$$+ \frac{R(\rho)}{2m}\sum_{i=0}^{t-1}\rho^{t-1-i}\|\mathbf{z}^i - \widehat{\boldsymbol{\theta}}\|^2 + \frac{C}{2m}\rho^{t+1} + C(\rho)\nu^2,$$

*for any $\mathbf{v} = [v_1^\top, \ldots, v_m^\top]^\top$, with feasible $v_1, \ldots, v_m$, where $\nu^2$ is defined in (29), and*

$$A(\rho) \triangleq \beta^2\alpha\rho + 3m\sqrt{m}\rho\left(\left(3 + \frac{2\rho}{1-\rho}\right)(2\gamma_\ell + 8\Psi^2(\bar{\mathcal{M}})\tau_\ell) + 2L + 8\Psi^2(\bar{\mathcal{M}})\tau_L\right),$$

$$B(\rho) \triangleq \frac{\beta^2\alpha\rho}{1+\beta} + \frac{3m\sqrt{m}\rho}{1+\beta}\left(2\gamma_\ell + 8\Psi^2(\bar{\mathcal{M}})\tau_\ell\right),$$

$$R(\rho) \triangleq 6m\sqrt{m}\rho\left(2\gamma_\ell + 8\Psi^2(\bar{\mathcal{M}})\tau_\ell\right), \tag{32}$$

$$C \triangleq 4R\sqrt{m}m\sum_{i=1}^{m}\mathcal{R}^*(\nabla f_i(\theta^\star)) + 3m\sqrt{m}(2\gamma_\ell + 8\Psi^2(\bar{\mathcal{M}}))\tau_\ell\|\widehat{\boldsymbol{\theta}} - \theta^\star\|^2,$$

$$C(\rho) \triangleq 9\sqrt{m}m\rho\left(\left(3 + \frac{2\rho}{1-\rho}\right)\tau_\ell + \tau_L\right).$$

*Proof.* See Appendix A.7.3. □

We can now combine the above intermediate results and prove convergence of Algorithm 1, stated in Theorem 3 below–Theorem 2 (in the main paper) follows as special case (see Remark 1).

To state the theorem, it is convenient to introduce the following quantities:

$$B \triangleq \boldsymbol{F}(\boldsymbol{\theta}^0) - \boldsymbol{F}(\widehat{\boldsymbol{\theta}}) + \frac{\mu}{16m}\|\mathbf{v}^0 - \widehat{\boldsymbol{\theta}}\|^2 + \frac{\beta\mu}{2m}\|\mathbf{z}^{-1} - \widehat{\boldsymbol{\theta}}\|^2 + \frac{\mu}{16m}\|\boldsymbol{\theta}^0\|^2_{\mathbf{I}-\mathbf{W}^2}, \tag{33}$$

$$\lambda \triangleq \frac{1 - \sqrt{\frac{1}{8\kappa}} + \left(1 - \sqrt{\frac{1}{8\kappa}}\right)\left(\frac{24\Psi^2(\bar{\mathcal{M}})\tau_L}{2L} - \frac{\sqrt{8\kappa}\tau(\rho)}{2L}\right) + \frac{\sqrt{8\kappa}\tau(\rho)}{L}}{1 - \frac{24\tau_L\Psi^2(\bar{\mathcal{M}})}{2L} - \frac{\sqrt{8\kappa}\tau(\rho)}{2L}} \tag{34}$$

where

$$\tau(\rho) \triangleq 3m\sqrt{m}\rho\left(\left(4 + \frac{2\rho}{1-\rho}\right)(2\gamma_\ell + 8\Psi^2(\bar{\mathcal{M}})\tau_\ell) + 3L + 12\Psi^2(\bar{\mathcal{M}})\tau_L\right). \tag{35}$$

Notice that the rate $\lambda$ in (34) matches that of the centralized PGA applied to the ERM (3) [1], with an improved dependence on $\kappa$, which gains the square root (typical of accelerated methods).

**Theorem 3** (Convergence of Algorithm 1). *Given the ERM* (3), *suppose* $(i)$ $f$ *satisfies the RSC/RSM conditions (Assumption 1); (ii)* $\mathcal{R}$ *is a decomposable regularizer with respect to a chosen pair of subspaces* $(\mathcal{M}, \bar{\mathcal{M}})$ *(Assumption 3); (iii)* $f_i$ *satisfies the local RSM condition (Assumption 2) for all* $i \in [m]$. *Let* $\{(\theta_i^t)_{i\in[m]}\}_{t\geq 1}$ *be the sequence generated by Algorithm 1, with tuning* $\alpha = 2L$ *and* $\beta = \sqrt{1/(8\kappa)}$ *and gossip matrix satisfying* $W \succeq 0$ *and Assumption 5, with* $\rho$ *such that*

$$\rho \leq \min\left\{\frac{56}{128}\beta, \frac{\kappa^{-1}}{384m\sqrt{m}\left(L^{-1}(16\gamma_\ell + 64\Psi^2(\bar{\mathcal{M}})\tau_\ell + 12\Psi^2(\bar{\mathcal{M}})\tau_L) + 3\right)}\right\}. \tag{36}$$

*If* $(\mathcal{M}, \bar{\mathcal{M}})$ *and RSC/RSM parameters* $(\mu, \tau_\mu)$ *and* $(L, \tau_L)$ *are such that*

$$\frac{\mu}{8} \geq 8\tau_\mu\Psi^2(\bar{\mathcal{M}}) + 24\tau_L\Psi^2(\bar{\mathcal{M}}), \tag{37}$$

*then, for any solution* $\widehat{\theta}$ *of* (3) *for which* $\mathcal{R}(\widehat{\theta}) = R$, *there holds*

$$\frac{1}{m}\left\|\boldsymbol{\theta}^{t+1} - \widehat{\boldsymbol{\theta}}\right\|^2 \leq \frac{8}{\mu}(\lambda + \rho)^{t+1}B$$

$$+ (\lambda + \rho)^{t+1}\frac{2\sqrt{2}\rho}{\sqrt{\mu L}}\left(4R\sqrt{m}\sum_{i=1}^m \mathcal{R}^*(\nabla f_i(\theta^\star)) + 3\sqrt{m}(2\gamma_\ell + 8\Psi^2(\bar{\mathcal{M}})\tau_\ell)\|\widehat{\boldsymbol{\theta}} - \boldsymbol{\theta}^\star\|^2\right)$$

$$+ \left(\frac{2\sqrt{2}}{(1 - \lambda - \rho)\sqrt{\mu L}}\left(9m\sqrt{m}\rho\left(\left(3 + \frac{2}{1-\rho}\right)\tau_\ell + \tau_L\right)\right) + \frac{4}{\mu}\tau_\mu\right)$$

$$\times 2\left(2\mathcal{R}(\Pi_{\mathcal{M}^\perp}(\theta^\star)) + 2\mathcal{R}(\theta^\star - \widehat{\theta}) + \Psi(\bar{\mathcal{M}})\|\theta^\star - \widehat{\theta}\|\right)^2 \tag{38}$$

*where* $B$ *and* $\lambda$ *are defined in (33) and (34), respectively.*

*Proof.* Combining Proposition 1 with Lemma 2 and Proposition 2 yields

$$\boldsymbol{F}(\mathbf{W}\boldsymbol{\theta}^{t+1}) - \boldsymbol{F}(\mathbf{W}\widehat{\boldsymbol{\theta}}) + \underbrace{\frac{\beta^2\alpha}{2m}}_{\triangleq q'_\theta}\|\boldsymbol{\theta}^{t+1}\|_{\mathbf{I}-\mathbf{W}^2}^2 + \underbrace{\left(\frac{\beta^2\alpha}{2m} - \frac{\beta A(\rho)}{2m} - \frac{\beta^2 24\tau_L\Psi^2(\bar{\mathcal{M}})}{2m}\right)}_{\triangleq q'_v}\|\mathbf{v}^{t+1} - \widehat{\boldsymbol{\theta}}\|^2$$

$$+ \underbrace{\left(\frac{\beta\mu}{2m} - \frac{\beta^3\alpha}{2m} - \frac{\beta}{2m}8\tau_\mu\Psi^2(\bar{\mathcal{M}}) - \frac{24\tau_L\beta^4}{2m}\Psi^2(\bar{\mathcal{M}}) - \frac{\beta A(\rho)}{2m}\right)}_{\triangleq q'_z}\|\mathbf{z}^t - \widehat{\boldsymbol{\theta}}\|^2 \leq$$

$$(1 - \beta)\left(\boldsymbol{F}(\mathbf{W}\boldsymbol{\theta}^t) - \boldsymbol{F}(\mathbf{W}\widehat{\boldsymbol{\theta}})\right) + \underbrace{\left((1-\beta)\frac{\beta^2\alpha}{2m} + \frac{B(\rho)\beta}{2m}\right)}_{\triangleq q''_\theta}\|\boldsymbol{\theta}^t\|_{\mathbf{I}-\mathbf{W}^2}^2$$

$$+ \underbrace{\left(\frac{(1-\beta)\beta^2\alpha}{2m} + \frac{24\tau_L\Psi^2(\bar{\mathcal{M}})}{2m}\beta^2(1-\beta)^2 + \frac{B(\rho)\beta^2}{2m}\right)}_{\triangleq q''_v}\|\mathbf{v}^t - \widehat{\boldsymbol{\theta}}\|^2$$

$$+ \underbrace{\frac{\beta R(\rho)}{2m}}_{\triangleq q''_z}\|\mathbf{z}^{t-1} - \widehat{\boldsymbol{\theta}}\|^2 + \underbrace{\frac{\beta R(\rho)}{2m}\sum_{i=0}^{t-2}\rho^{t-i-1}\|\mathbf{z}^i - \widehat{\boldsymbol{\theta}}\|^2 + \frac{\beta C}{2m}\rho^{t+1} + \left(\beta C(\rho) + \frac{\beta}{2}(\tau_\mu + 9\tau_L\beta)\right)\nu^2}_{\triangleq \delta^t}. \tag{39}$$

Introducing

$$B^{t+1} \triangleq \boldsymbol{F}(\mathbf{W}\boldsymbol{\theta}^{t+1}) - \boldsymbol{F}(\widehat{\boldsymbol{\theta}}) + q_\theta'\|\boldsymbol{\theta}^{t+1}\|_{\mathbf{I}-\mathbf{W}^2}^2 + q_v'\|\mathbf{v}^{t+1} - \widehat{\boldsymbol{\theta}}\|^2 + q_z'\|\mathbf{z}^t - \widehat{\boldsymbol{\theta}}\|^2$$
$$= \boldsymbol{g}(\boldsymbol{\theta}^{t+1}) + q_v'\|\mathbf{v}^{t+1} - \widehat{\boldsymbol{\theta}}\|^2 + q_z'\|\mathbf{z}^t - \widehat{\boldsymbol{\theta}}\|^2, \tag{40}$$

and

$$\bar{\lambda} = \max\left\{1 - \beta, \frac{q_\theta''}{q_\theta'}, \frac{q_v''}{q_v'}, \frac{q_z''}{q_z'}\right\}, \quad \text{for} \quad q_z', q_\theta', q_v' > 0, \tag{41}$$

(39) yields

$$B^{t+1} \leq \bar{\lambda}B^t + \delta^t.$$

We will enforce shortly conditions on the system parameters via $q_\theta'$, $q_v'$, $q_z'$ and $q_\theta''$, $q_v''$, $q_z''$ so that $\bar{\lambda} \in (0, 1)$. Hence, hereafter, we will assume $\bar{\lambda} \in (0, 1)$.

Using $q_z'' \leq \bar{\lambda} q_z'$, due to (41), and the expression of $\delta^t$ (see (39)), we can write

$$B^{t+1} \leq \bar{\lambda}B^t + q_z'\bar{\lambda}\sum_{i=0}^{t-2}\rho^{t-i-1}\|\mathbf{z}^i - \widehat{\boldsymbol{\theta}}\|^2 + \frac{\beta C}{2m}\rho^{t+1} + \left(\beta C(\rho) + \frac{\beta}{2}\left(\tau_\mu + 9\tau_L\beta\right)\right)\nu^2. \tag{42}$$

To study the dynamics of (42), define the sequence $\{S^t\}$ as

$$S^t = \rho\, S^{t-1} + \sqrt{\rho\bar{\lambda}}B^{t-1}, \quad t = 1, 2\ldots; \quad \text{and} \quad S^0 = 0.$$

Notice that

$$S^t = \sqrt{\rho\bar{\lambda}}\sum_{i=0}^{t-1}\rho^{t-1-i}B^i, \quad t = 1, 2, \ldots.$$

Consequently, we can write: for any $t = 0, 1, \ldots$,

$$B^{t+1} \leq \bar{\lambda}B^t + \sqrt{\rho\bar{\lambda}}S^t + \frac{\beta C}{2m}\rho^{t+1} + \left(\beta C(\rho) + \frac{\beta}{2}(\tau_\mu + \tau_L\beta)\right)\nu^2,$$
$$S^{t+1} \leq \rho S^t + \sqrt{\bar{\lambda}\rho}B^t;$$

and

$$\left\|\begin{matrix}B^{t+1}\\S^{t+1}\end{matrix}\right\| \leq \left\|\begin{pmatrix}\bar{\lambda} & \sqrt{\bar{\lambda}\rho}\\\sqrt{\bar{\lambda}\rho} & \rho\end{pmatrix}\right\|_2\left\|\begin{matrix}B^t\\S^t\end{matrix}\right\| + \frac{\beta C}{2m}\rho^t + \left(\beta C(\rho) + \frac{\beta}{2}(\tau_\mu + 9\tau_L\beta)\right)\nu^2.$$

The matrix above has spectral norm equal to $\bar{\lambda} + \rho$. Consequently, under

$$\bar{\lambda} + \rho < 1, \tag{43}$$

by telescoping back to $t = 0$ while using $S^0 = 0$, we obtain

$$B^{t+1} \leq \left(\bar{\lambda} + \rho\right)^{t+1} B^0 + \frac{\beta C}{2m}\sum_{i=0}^{t}\rho^{i+1}(\bar{\lambda} + \rho)^{t-i} + \frac{1}{1 - \bar{\lambda} - \rho}\left(\beta C(\rho) + \frac{\beta}{2}(\tau_\mu + 9\tau_L\beta)\right)\nu^2$$

$$\leq (\bar{\lambda} + \rho)^{t+1}B^0 + \frac{\beta C}{2m}(\bar{\lambda} + \rho)^{t+1}\frac{\rho}{\bar{\lambda}} + \frac{1}{1 - \bar{\lambda} - \rho}\left(\beta C(\rho) + \frac{\beta}{2}(\tau_\mu + 9\tau_L\beta)\right)\nu^2. \tag{44}$$

Using (40), we can lower bound $B^{t+1}$ as

$$B^{t+1} \geq \boldsymbol{g}(\boldsymbol{\theta}^{t+1}) - \boldsymbol{g}(\widehat{\boldsymbol{\theta}})$$

$$\overset{(a)}{\geq} \frac{\mu}{4m}\|\boldsymbol{\theta}^{t+1} - \widehat{\boldsymbol{\theta}}\|_{\mathbf{W}^2}^2 + \frac{\beta^2\alpha}{2m}\|\boldsymbol{\theta}^{t+1} - \widehat{\boldsymbol{\theta}}\|_{\mathbf{I}-\mathbf{W}^2}^2 - \varepsilon_\ell(\boldsymbol{\theta}^{t+1}, \widehat{\boldsymbol{\theta}})$$

$$\overset{(30)}{\geq} \frac{\beta^2\alpha}{2m}\|\boldsymbol{\theta}^{t+1} - \widehat{\boldsymbol{\theta}}\|^2 + \left(\frac{\mu}{4m} - \frac{\tau_\mu}{2m}8\Psi^2(\bar{\mathcal{M}}) - \frac{\beta^2\alpha}{2m}\right)\|\boldsymbol{\theta}^{t+1} - \widehat{\boldsymbol{\theta}}\|_{\mathbf{W}^2}^2 - \frac{\tau_\mu}{2}\nu^2, \tag{45}$$

where (a) follows from (25a) and $\langle \nabla g(\widehat{\boldsymbol{\theta}}), \boldsymbol{\theta}^{t+1} - \widehat{\boldsymbol{\theta}} \rangle \geq 0$.

Chaining (44) and (45), we finally obtain

$$\|\boldsymbol{\theta}^{t+1} - \widehat{\boldsymbol{\theta}}\|^2 + \left( \frac{\mu}{2\beta^2\alpha} - \frac{\tau_\mu}{\beta^2\alpha^2} 8\Psi^2(\bar{\mathcal{M}}) - 1 \right) \|\boldsymbol{\theta}^{t+1} - \widehat{\boldsymbol{\theta}}\|^2_{\mathbf{W}^2} \leq \frac{2m}{\beta^2\alpha}(\bar{\lambda} + \rho)^{t+1} V^0$$

$$+ (\bar{\lambda} + \rho)^{t+1} \frac{\rho}{\bar{\lambda}\beta\alpha} \left( 4R\sqrt{m}m \sum_{i=1}^m \mathcal{R}^*(\nabla f_i(\theta^\star)) + 3m\sqrt{m}(2\gamma_\ell + 8\Psi^2(\bar{\mathcal{M}})\tau_\ell)\|\widehat{\boldsymbol{\theta}} - \boldsymbol{\theta}^\star\|^2 \right)$$

$$+ \frac{2m}{\beta\alpha(1 - \bar{\lambda} - \rho)} \left( 9m\sqrt{m}\rho \left( \left( 3 + \frac{2}{1-\rho} \right) \tau_\ell + \tau_L \right) + \frac{\tau_\mu + 9\tau_L}{2} \right) \nu^2 + \frac{m}{\beta^2\alpha} \frac{\tau_\mu}{2} \nu^2.$$
(46)

We proceed now to establish sufficient conditions for $\bar{\lambda}$ in (41) to be strictly less than one as well as for (43) to hold, which have been subsumed in the above derivations.

Substituting the expressions of $q'_\theta$, $q'_v$, $q'_z$ and $q''_\theta$, $q''_v$, $q''_z$ (as in (39)), in (41), we have

$$\frac{\beta^2\alpha}{2m} > 0,$$

$$\frac{\beta^2\alpha}{2m} - \frac{\beta A(\rho)}{2m} - \frac{\beta^2 24\tau_L \Psi^2(\bar{\mathcal{M}})}{2m} > 0,$$

$$\frac{\beta\mu}{2m} - \frac{\beta^3\alpha}{2m} - \frac{\beta}{2m} 8\tau_\mu \Psi^2(\bar{\mathcal{M}}) - \frac{24\tau_L \beta^4 \Psi^2(\bar{\mathcal{M}})}{2m} - \frac{\beta A(\rho)}{2m} > 0,$$

$$\bar{\lambda} = \max \left\{ 1 - \beta, \frac{(1-\beta)\beta^2\alpha + B(\rho)\beta}{\beta^2\alpha}, \frac{(1-\beta)\beta^2\alpha + 24\tau_L \Psi^2(\bar{\mathcal{M}})\beta^2(1-\beta)^2 + B(\rho)\beta^2}{\beta^2\alpha - \beta A(\rho) - \beta^2 24\tau_L \Psi^2(\bar{\mathcal{M}})}, \right.$$

$$\left. \frac{\beta R(\rho)}{\beta\mu - \beta^3\alpha - \beta 8\tau_\mu \Psi^2(\bar{\mathcal{M}}) - 24\tau_L \beta^4 \Psi^2(\bar{\mathcal{M}}) - \beta A(\rho)} \right\}.$$

Sufficient conditions are

$$\beta\alpha - A(\rho) - \beta 24\tau_L \Psi^2(\bar{\mathcal{M}}) > 0, \tag{47a}$$

$$\mu - \beta^2\alpha - 8(\tau_\mu + 24\beta^3\tau_L)\Psi^2(\bar{\mathcal{M}}) - A(\rho) > 0, \tag{47b}$$

$$\bar{\lambda} = \max \left\{ 1 - \beta + \frac{(1-\beta)(\beta A(\rho) + \beta^2 24\tau_L \Psi^2(\bar{\mathcal{M}})) + 24\Psi^2(\bar{\mathcal{M}})\beta^2(1-\beta)^2 + B(\rho)\beta}{\beta^2\alpha - \beta A(\rho) - \beta^2 24\tau_L \Psi^2(\bar{\mathcal{M}})}, \right.$$

$$\left. \frac{R(\rho)}{\mu - \beta^2\alpha - 8\psi^2(\bar{\mathcal{M}}) - 24\tau_L \beta^3 \Psi^2(\bar{\mathcal{M}}) - \beta A(\rho)} \right\}. \tag{47c}$$

By enforcing

$$\frac{R(\rho)}{\mu - \beta^2\alpha - 8\tau_\mu \Psi^2(\bar{\mathcal{M}}) - 24\tau_L \beta^3 \Psi^2(\bar{\mathcal{M}}) - A(\rho)} \leq 1 - \beta, \tag{48}$$

we have

$$\bar{\lambda} \leq 1 - \beta + \frac{(1-\beta)A(\rho) + B(\rho) + 48\beta(1-\beta)\tau_L \Psi^2(\bar{\mathcal{M}})}{\beta\alpha - A(\rho) - 24\beta\tau_L \Psi^2(\bar{\mathcal{M}})}. \tag{49}$$

For (48) to hold it is sufficient that

$$\mu - \beta^2\alpha - 8\tau_\mu \Psi^2(\bar{\mathcal{M}}) - 24\tau_L \beta^3 \Psi^2(\bar{\mathcal{M}}) - A(\rho) \geq \frac{\mu}{2}, \tag{50a}$$

$$R(\rho) \leq \frac{\mu}{2}(1 - \beta). \tag{50b}$$

Notice that (50a) also implies (47b).

Choose

$$\beta^2 = \frac{\mu}{8L}. \tag{51}$$

Sufficient conditions for (50) to hold are

$$\frac{\mu}{4} \geq \max \left\{ 8\tau_\mu \Psi^2(\bar{\mathcal{M}}) + 24\tau_L \Psi^2(\bar{\mathcal{M}}), A(\rho), R(\rho) \right\}. \tag{52}$$

Using now the expression of $A(\rho)$ and $R(\rho)$ (see (32)) and enforcing

$$\rho \leq \frac{1}{2}, \tag{53}$$

(52) is implied by

$$\beta^2 \alpha \rho + 3m\sqrt{m}\rho \left(7(2\gamma_\ell + 8\Psi^2(\bar{\mathcal{M}})\tau_\ell) + 2L + 8\Psi^2(\bar{\mathcal{M}})\tau_L\right) \leq \frac{\mu}{4}, \tag{54a}$$

$$8\tau_\mu \Psi^2(\bar{\mathcal{M}}) + 24\tau_L \Psi^2(\bar{\mathcal{M}}) \leq \frac{\mu}{4}. \tag{54b}$$

The following additional condition on $\rho$ is sufficient for (54a):

$$\rho \leq \frac{\mu}{12m\sqrt{m}\left(14\gamma_\ell + 56\Psi^2(\bar{\mathcal{M}})\tau_\ell + 2L + 8\Psi^2(\bar{\mathcal{M}})\tau_L\right) + \mu}. \tag{55}$$

Observe that under (52), $\beta$ given by (51) and $\alpha \geq 2L$, the LHS of (47a) is lower bounded by $\sqrt{\frac{\mu L}{2}} - \frac{\mu}{4} - \sqrt{\frac{\mu}{2L}}\frac{\mu}{8}$; therefore (47a) is fulfilled, for any values of $\mu$ and $L$.

It remains to show that (43) holds. We claim that, using therein the upper bound of $\bar{\lambda}$ in (49), with $\beta$ given by (51), (43) holds under

$$24\tau_L \Psi^2(\bar{\mathcal{M}}) \leq \frac{\mu}{8}, \tag{56}$$

as long as $\rho$ satisfies (36). In fact, first, notice that using definition of the parameters $A(\rho)$ and $B(\rho)$ it holds

$$\bar{\lambda} \leq \lambda,$$

where $\lambda$ is given by (34). Then, substituting (36) in (35) and using $\tau(\rho) \leq \frac{\mu}{128}$, we deduce

$$\bar{\lambda} \leq \lambda \leq 1 - \frac{7}{8}\beta + \beta^2(1-\beta) \leq 1 - \frac{7 - \sqrt{8}}{8}\beta,$$

which proves (43).

Finally, notice that (36) is sufficient for (53) and (55).

To summarize, we proved (46), under (36) and (54b), (56), with (54b), (56) implied by (37). The final convergence as in (38) follows from (46), lower bounding the LHS by $\frac{1}{m}\|\boldsymbol{\theta}^{t+1} - \widehat{\boldsymbol{\theta}}\|^2$ and using (52) and $\bar{\lambda}^{-1} \leq 2$. $\qquad \square$

**Remark 1** (Theorem 2 as special case of Theorem 3). *Using $\tau(\rho) \leq \frac{\mu}{128}$, we can bound $\lambda$ in (34) as*

$$\lambda \leq 1 - \sqrt{\frac{1}{8\kappa}} + \left(1 - \sqrt{\frac{1}{8\kappa}}\right)\frac{48\tau_L\Psi^2(\bar{\mathcal{M}})}{\frac{127}{64}L - 24\tau_L\Psi^2(\bar{\mathcal{M}})} + \frac{\sqrt{8\mu L}/64}{\frac{127}{64}L - 24\tau_L\Psi^2(\bar{\mathcal{M}})}.$$

*Under $24\tau_L\Psi^2(\bar{\mathcal{M}}) \leq \frac{\sqrt{2}\mu}{14}$ and $\rho \leq \frac{\sqrt{\kappa^{-1}}}{18}$, one gets*

$$\lambda + \rho \leq 1 - \sqrt{\frac{1}{32\kappa}}.$$

*This yields Theorem 2.*

## A.5 Proof of Theorem 1

Theorem 1 is proved as special case of the theorem below–Remark 2 to follow elaborate on the connection.

**Theorem 4.** *Given the ERM (3), suppose (i) $f$ satisfies the RSC/RSM conditions (Assumption 1); (ii) $\mathcal{R}$ is a decomposable regularizer with respect to a chosen pair of subspaces $(\mathcal{M}, \bar{\mathcal{M}})$ (Assumption 3); (iii) $(\bar{\mathcal{M}}, \bar{\mathcal{M}})$ and the RSC/RSM parameters $(\mu, \tau_\mu)$ and $(L, \tau_L)$ are such that*

$$\frac{\mu}{8} \geq 8\tau_\mu \Psi^2(\bar{\mathcal{M}}) + 24\tau_L \Psi^2(\bar{\mathcal{M}}). \tag{57}$$

*Let $\{\theta^t\}_{t \geq 1}$ be the sequence generated by the algorithm in (9) with tuning $\alpha = 2L$ and $\beta = \sqrt{1/(8\kappa)}$. Then, it holds*

$$\|\theta^{t+1} - \widehat{\theta}\|^2 \leq \frac{8}{\mu} \lambda^{t+1} \left( f(\theta^0) - f(\widehat{\theta}) + \frac{\mu}{8}\|v^0 - \widehat{\theta}\|^2 \right) \tag{58}$$

$$+ \frac{4}{\mu} \left( \frac{1}{1-\lambda}(\beta\tau_\mu + 9\tau_L\beta^2) + \tau_\mu \right) \times 2 \left( 2\mathcal{R}(\Pi_{\mathcal{M}^\perp}(\theta^\star)) + 2\mathcal{R}(\theta^\star - \widehat{\theta}) + \Psi(\bar{\mathcal{M}})\|\widehat{\theta} - \theta^\star\| \right)^2,$$

*with*

$$\lambda \triangleq \frac{1 - \sqrt{\frac{1}{8\kappa}} + \left(1 - \sqrt{\frac{1}{8\kappa}}\right)\left(\frac{24\Psi^2(\bar{\mathcal{M}})\tau_L}{2L}\right)}{1 - \frac{24\tau_L\Psi^2(\bar{\mathcal{M}})}{2L}}.$$

*Proof.* We apply Lemma 1, with $g = f$; we can write

$$f(\theta^{t+1}) - f(\widehat{\theta}) + \frac{\beta^2\alpha}{2}\|v^{t+1} - \widehat{\theta}\|^2 \leq (1-\beta)(f(\theta^t) - f(\widehat{\theta})) + \frac{(1-\beta)\beta^2\alpha}{2}\|v^t - \widehat{\theta}\|^2$$

$$- \left(\frac{\beta\mu}{4} - \frac{\beta^3\alpha}{2}\right)\|\widehat{\theta} - z^t\|^2 + \beta\frac{\tau_\mu}{2}\mathcal{R}(\widehat{\theta} - z^t) + \varepsilon_u\frac{\tau_L}{2}\mathcal{R}^2(\theta^{t+1} - z^t).$$

Using Lemma 2 we have

$$\frac{\tau_\mu}{2}\mathcal{R}^2(\widehat{\theta} - z^t) \leq \frac{\tau_\mu}{2}\left(8\Psi^2(\bar{\mathcal{M}})\|\widehat{\theta} - z^t\|^2 + \nu^2\right),$$

$$\frac{\tau_L}{2}\mathcal{R}^2(\theta^{t+1} - z^t) \leq \frac{\tau_L}{2}\left(24\beta^2\Psi^2(\bar{\mathcal{M}})\left(\|v^{t+1} - \widehat{\theta}\|^2 + (1-\beta)^2\|v^t - \widehat{\theta}\|^2 + \beta^2\|z^t - \widehat{\theta}\|^2\right) + 9\beta^2\nu^2\right).$$

Therefore,

$$f(x^{t+1}) - f(\widehat{\theta}) + \left(\frac{\beta^2\alpha}{2} - \frac{24\tau_L}{2}\beta^2\Psi^2(\bar{\mathcal{M}})\right)\|v^{t+1} - \widehat{\theta}\|^2$$

$$\leq (1-\beta)(f(x^t) - f(\widehat{\theta})) + \left(\frac{(1-\beta)\beta^2\alpha}{2} + \frac{24\tau_L\beta^2}{2}(1-\beta)^2\Psi^2(\bar{\mathcal{M}})\right)\|v^t - \widehat{\theta}\|^2 + \left(\frac{\beta\tau_\mu}{2} + \frac{9\tau_L}{2}\beta^2\right)\nu^2$$

$$- \left(\frac{\beta\mu}{4} - \frac{\beta^3\alpha}{2} - \frac{24\tau_L\beta^4\Psi^2(\bar{\mathcal{M}})}{2} - \frac{\beta\tau_\mu}{2}8\Psi^2(\bar{\mathcal{M}})\right)\|z^t - \widehat{\theta}\|^2.$$

Enforcing

$$\beta^2\alpha + 24\tau_L\beta^3\Psi^2(\bar{\mathcal{M}}) + 8\Psi^2(\bar{\mathcal{M}})\tau_\mu \leq \frac{\mu}{2}, \tag{59}$$

$$(1-\beta)\beta^2\alpha + 24\tau_L\beta^2(1-\beta)^2\Psi^2(\bar{\mathcal{M}}) \leq \bar{\lambda}\underbrace{\left(\beta^2\alpha - 24\tau_L\beta^2\Psi^2(\bar{\mathcal{M}})\right)}_{\triangleq 2q_v}, \tag{60}$$

for some $\bar{\lambda} \in [1-\beta, 1)$ (to be determined), yields

$$f(\theta^{t+1}) - f(\widehat{\theta}) + q_v\|v^{t+1} - \widehat{\theta}\|^2 \leq \bar{\lambda}\left(f(x^t) - f(\widehat{\theta}) + q_v\|v^t - \widehat{\theta}\|^2\right) + \left(\frac{\beta\tau_\mu}{2} + \frac{9\tau_L}{2}\beta^2\right)\nu^2,$$

whereby telescoping we have: for $t = 0, 1, \ldots$,

$$f(\theta^{t+1}) - f(\widehat{\theta}) \leq \bar{\lambda}^{t+1}\left(f(\theta^0) - f(\widehat{\theta}) + q_v\|v^0 - \widehat{\theta}\|^2\right) + \frac{1}{1-\bar{\lambda}}\left(\frac{\beta\tau_\mu}{2} + \frac{9\tau_L}{2}\beta^2\right)\nu^2. \tag{61}$$

Invoking the RSC of $f$ and using optimimality of $\widehat{\theta}$, we can write

$$f(\theta^{t+1}) - f(\widehat{\theta}) \geq \frac{\mu}{4}\|\theta^{t+1} - \widehat{\theta}\|^2 - \frac{\tau_\mu}{2}\mathcal{R}^2(\theta^{t+1} - \widehat{\theta})$$

$$\overset{(30)}{\geq} \frac{\mu}{4}\|\theta^{t+1} - \widehat{\theta}\|^2 - \frac{\tau_\mu}{2}\left(8\Psi^2(\bar{\mathcal{M}})\|\theta^{t+1} - \widehat{\theta}\|^2 + \nu^2\right). \tag{62}$$

Chaining (61) and (62) under

$$8\tau_\mu \Psi^2(\bar{\mathcal{M}}) \leq \frac{\mu}{4}, \tag{63}$$

and using $q_v \leq \frac{\beta^2\alpha}{2}$, we obtain

$$\|\theta^{t+1} - \widehat{\theta}\|^2 \leq \frac{8}{\mu}\lambda^{t+1}\left(f(\theta^0) - f(\widehat{\theta}) + \frac{\beta^2\alpha}{2}\|v^0 - \widehat{\theta}\|^2\right) + \frac{8}{\mu}\left(\frac{1}{1-\lambda}\left(\frac{\beta\tau_\mu}{2} + \frac{9\tau_L}{2}\beta^2\right) + \frac{\tau_\mu}{2}\right)\nu^2.$$

Observe that the smallest $\bar{\lambda}$ one can choose while satisfying (60) is

$$\bar{\lambda} = \max\left\{1-\beta, \frac{(1-\beta)\alpha + 24\tau_L(1-\beta)^2\Psi^2(\bar{\mathcal{M}})}{\alpha - 24\tau_L\Psi^2(\bar{\mathcal{M}})}\right\},$$

and thus

$$\bar{\lambda} = 1 - \beta + \frac{24\tau_L((1-\beta)^2 + (1-\beta))\Psi^2(\bar{\mathcal{M}})}{\alpha - 24\tau_L\Psi^2(\bar{\mathcal{M}})} \leq \frac{1 - \beta + (1-\beta)\left(\frac{24\Psi^2(\bar{\mathcal{M}})\tau_L}{2L}\right)}{1 - \frac{24\tau_L\Psi^2(\bar{\mathcal{M}})}{2L}}.$$

To summarize, the above convergence has been achieved under (63) and (59), implied by (57). The upper bound on the rate as above is compliant with (60). Notice that (57) also guarantees that the RHS of (60) is strictly positive. □

**Remark 2.** *By further assuming that* $24\Psi^2(\bar{\mathcal{M}})\tau_L \leq \frac{\sqrt{2}\mu}{14}$ *it holds that*

$$\lambda \leq 1 - \sqrt{\frac{1}{16\kappa}}.$$

*This yields the result in Theorem 1.*

### A.6 Proofs of Corollaries 1-4

### A.6.1 Proof of Corollary 1

In the setting of Corollary 1 the following holds

$$\Psi^2(\bar{\mathcal{M}}) \leq \left(\frac{\log d}{N}\right)^{-\frac{q}{2}} R_q, \qquad \|\Pi_{\mathcal{M}^\perp}(\theta^\star)\|_1 \leq \left(\frac{\log d}{N}\right)^{\frac{1-q}{2}} R_q, \tag{64a}$$

$$\gamma_\ell = (2m+1)L, \qquad \tau_\ell = \eta_\Sigma c_0 \frac{\log d}{n}, \tag{64b}$$

$$\mu = \lambda_{\min}(\Sigma), \qquad L = \lambda_{\max}(\Sigma), \tag{64c}$$

$$\tau_\mu = \tau_L = \eta_\Sigma c_0 \frac{\log d}{N}, \qquad \max_{i\in[m]}\|\nabla f_i(\theta^\star)\|_1 \leq c_1\sqrt{m\eta_\sigma\sigma^2\frac{log(md)}{N}}, \tag{64d}$$

with probability at least

$$1 - c_2\exp\left(-c_3N + \log(m)\right),$$

for some universal constants $c_0$, $c_1$, $c_2$, $c_3 > 0$ and all $q \in [0,1]$. The proof of the above result follows from [26], [11] with minor modifications, hence it is omitted.

Invoking Theorem 3 given that

$$R_q c_4 \eta_\Sigma \left(\frac{\log d}{N}\right)^{1-\frac{q}{2}} \leq \mu,$$

and the network

$$\rho \leq \frac{\kappa^{-1}}{384m\sqrt{m}(L^{-1}(16(2m+1)) + c_5(m+1)\mu + 3)},$$

for some universal constants $C_4$, $c_5 > 0$, with the parameters (64) yields

$$\frac{1}{m}\|\boldsymbol{\theta}^{t+1} - \widehat{\boldsymbol{\theta}}\|^2 \leq \frac{8}{\mu}\left(1 - \sqrt{\frac{1}{32\kappa}}\right)^{t+1}\left(f(0) - f(\widehat{\theta}) + \frac{\mu}{16m}\|\widehat{\boldsymbol{\theta}}\|^2 + \frac{\beta\mu}{2m}\|\widehat{\boldsymbol{\theta}}\|^2\right)$$

$$+ \frac{2\sqrt{2}}{\mu}\left(1 - \sqrt{\frac{1}{32\kappa}}\right)^{t+1}\frac{\kappa^{-3/2}}{96m\sqrt{m}\,(16(2m+1))} \times$$

$$\left(4R\sqrt{m}m\sqrt{c_2 m\eta_\Sigma\sigma^2\frac{\log(md)}{N}} + 3\sqrt{m}(2(2m+1)L + 8\mu)\|\theta^\star - \widehat{\theta}\|^2\right)$$

$$+ 24\left(\frac{2\sqrt{2}}{\mu}\frac{\kappa^{-3/2}}{96m\sqrt{m}((16(2m+1)))}9m\sqrt{m}\left(c_0\eta_\Sigma\frac{\log d}{N}\,(7m+1)\right) + \frac{4}{\mu}c_0\eta_\Sigma\frac{\log d}{N}\right) \times$$

$$\times\left(R_q^2\left(\frac{\log d}{N}\right)^{1-q} + \mathcal{R}^2(\theta^\star - \widehat{\theta}) + \left(\frac{\log d}{N}\right)^{-\frac{q}{2}}R_q\|\theta^\star - \widehat{\theta}\|^2\right),$$

whereby noticing that $\kappa^{-1} \leq 1$ and invoking Lemma 5 in the supplementary material of [1] which states

$$\mathcal{R}(\widehat{\theta} - \theta^\star) \leq 2\Psi(\bar{\mathcal{M}})\|\widehat{\theta} - \theta^\star\| + \mathcal{R}\left(\Pi_{\mathcal{M}^\perp}(\theta^\star)\right)$$

yields the desired result. $\qquad\square$

### A.6.2   Proof of Corollary 2

In the setting of Corollary 1 we have that for $q \in [0,1]$

$$\Psi^2(\bar{\mathcal{M}}) \leq R_q\left(\frac{p}{N}\right)^{-\frac{q}{2}}, \qquad\qquad \|\Pi_{\mathcal{M}^\perp}(\Theta^\star)\|_1 \leq R_q\left(\frac{p}{N}\right)^{\frac{1-q}{2}},$$

$$\gamma_\ell = (2m+1)L \qquad\qquad\qquad\qquad \tau_\ell = c_0\eta_\Sigma\frac{p}{n},$$

$$\mu = \lambda_{\min}(\Sigma) \qquad\qquad\qquad\qquad\quad L = \lambda_{\max}(\Sigma),$$

$$\tau_\mu = \tau_L = c_0\eta_\Sigma\frac{\log d}{N}, \qquad\qquad \max_{i\in[m]}\|\nabla f_i(\Theta^\star)\| \leq c_1\sigma\eta_\Sigma^{1/2}m\sqrt{\frac{p}{N}}.$$

The results above follow from small modifications of [1], [29], [26] and applying the union bound. Then, the desired result follows by taking the same steps as in the proof of Corollary 1. $\qquad\square$

### A.6.3   Proof of Corollary 3

In the setting of Corollary 3 we have that for $q \in [0,1]$

$$\Psi^2(\bar{\mathcal{M}}) \leq \left(\frac{\log d}{N}\right)^{-\frac{q}{2}}R_q, \qquad\qquad \|\Pi_{\mathcal{M}^\perp}(\theta^\star)\|_1 \leq \left(\frac{\log d}{N}\right)^{\frac{1-q}{2}}R_q,$$

$$\gamma_\ell = \frac{\lambda_{\max}(\Sigma)}{2} + \frac{m\lambda_{\min}(\Sigma)}{2}, \qquad\qquad \tau_\ell = c_0\max\left\{\frac{\tau^4}{\lambda_{\min}(\Sigma)}, \lambda_{\min}(\Sigma)\right\}\frac{\log d}{n},$$

$$\mu = \lambda_{\min}(\Sigma), \qquad\qquad\qquad\qquad\qquad L = \frac{\mu}{2} + \frac{\lambda_{\max}(\Sigma)}{2},$$

$$\tau_\mu = \tau_L = c_0\max\left\{\frac{\tau^4}{\lambda_{\min}(\Sigma)}, \lambda_{\min}(\Sigma)\right\}\frac{\log d}{N}, \quad \max_{i\in[m]}\|\nabla f_i(\theta^\star)\|_\infty \leq Cc_1\sqrt{\frac{\sigma m\log(md)}{N}},$$

with the probability given in the Corollary's statement. The above follow from small modifications of the statistical results in [17], [11], and using the union bound. Then, following the same steps as in the proof of Corollary 1 yields the desired result. $\qquad\square$

### A.6.4 Proof of Corollary 4

The matrix completion case deviates slightly from the other cases as the empirical risk does not fulfill exactly Assumptions 1 and 2, but a slightly modified version [1]. Observe that this is consistent with the results obtained for plain projected gradient descent in [1]. For the described model it holds that for all $i \in [m]$ with probability at least

$$1 - \exp\left(-p \log p\right) - c_1 m \exp(-p \log p),$$

and for all $V \in \mathbb{R}^{p \times p}$ and $V_i \in \mathbb{R}^{p \times p}$, it holds

$$\left\langle \operatorname{vec}(V), \frac{XX^\top}{N} \operatorname{vec}(V) \right\rangle \leq \|V\|_F^2 + c_2 p \|V\|_\infty \|V\|_1 \sqrt{\frac{p \log p}{N}} + c_3 \left( p \|V\|_\infty \sqrt{\frac{p \log p}{N}} \right)^2,$$

(65a)

$$\left\langle \operatorname{vec}(V), \frac{XX^\top}{N} \operatorname{vec}(V) \right\rangle \geq \|V\|_F^2 - c_2 p \|V\|_\infty \|V\|_1 \sqrt{\frac{p \log p}{N}} + c_3 \left( p \|V\|_\infty \sqrt{\frac{p \log p}{N}} \right)^2,$$

(65b)

$$\left\langle \operatorname{vec}(V_i), \frac{X_{\mathcal{S}_i} X_{\mathcal{S}_i}^\top}{n} \operatorname{vec}(V_i) \right\rangle \leq \|V_i\|_F^2 + c_2 pm \|V_i\|_\infty \|V_i\|_1 \sqrt{\frac{p \log p}{N}} + c_3 \left( pm \|V\|_\infty \sqrt{\frac{p \log p}{N}} \right)^2,$$

(65c)

$$\left\langle \operatorname{vec}(V_i), \frac{X_{\mathcal{S}_i} X_{\mathcal{S}_i}^\top}{n} \operatorname{vec}(V_i) \right\rangle \geq \|V_i\|_F^2 - c_2 pm \|V_i\|_\infty \|V_i\|_1 \sqrt{\frac{p \log p}{N}} - c_3 \left( pm \|V\|_\infty \sqrt{\frac{p \log p}{N}} \right)^2,$$

(65d)

where the statements involving $X$ are a restatement of Proposition 2 in the supplementary material of [1] and the remaining can be established by combining the proof of Proposition 2 in [1], the strategy employed to establish the RSM in [26] and the union bound. By setting $V = \Theta_1 - \Theta_2$ such that both $\Theta_1, \Theta_2 \in \Omega$ one can establish

$$f(\Theta_1) - f(\Theta_2) - \langle \nabla f(\Theta_2), \Theta_1 - \Theta_2 \rangle \leq \|\Theta_1 - \Theta_2\|_F^2$$
$$+ c_2 p \|\Theta_1 - \Theta_2\|_\infty \|\Theta_1 - \Theta_2\|_1 \sqrt{\frac{p \log p}{N}} + c_3 \left( p \|\Theta_1 - \Theta_2\|_\infty \sqrt{\frac{p \log p}{N}}, \right)^2$$

and an analogous lower bound. Observe that due to belonging to $\Omega$ it holds that

$$\|\Theta_1 - \Theta_2\|_\infty \leq 2 \frac{\omega}{p},$$

implying that the RSC and RSS hold upto a tolerance $c_3 \omega^2 \frac{p \log p}{N}$. For such a model, the procedure to establish convergence is identical to that in the proof of Theorem 3 with the difference that the additional tolerance needs to be absorbed with the misspecification error terms ($\nu^2$).

The remaining terms are given by

$$\Psi^2(\bar{\mathcal{M}}) \leq R_q \omega^{-q} \left( \frac{p \log p}{N} \right)^{-\frac{q}{2}},$$

$$\|\Pi_{\mathcal{M}^\perp}(\Theta^\star)\|_1 \leq R_q \omega^{1-q} \left( \frac{p \log p}{N} \right)^{\frac{1-q}{2}},$$

$$\max_{i \in [m]} \mathcal{R}^* \left( \nabla f_i(\Theta^\star) \right) \leq c_1 m \sigma \sqrt{\frac{p \log p}{N}},$$

which can be found in [1],[29].

### A.7 Supporting results in the proof of Theorem 3

#### A.7.1 Proof of Proposition 1

From (25b) it follows

$$
\begin{aligned}
\boldsymbol{g}(\boldsymbol{\theta}^{t+1}) &\leq \boldsymbol{g}(\mathbf{w}^t) + \langle \nabla \boldsymbol{g}(\mathbf{w}^t), \boldsymbol{\theta}^{t+1} - \mathbf{w}^t \rangle + \frac{L}{m}\|\boldsymbol{\theta}^{t+1} - \mathbf{w}^t\|_{\mathbf{W}^2}^2 + \varepsilon_u(\boldsymbol{\theta}^{t+1}, \mathbf{w}^t) \\
&\quad + \frac{\beta^2 \alpha}{2m}\|\boldsymbol{\theta}^{t+1} - \mathbf{w}^t\|_{\mathbf{I}-\mathbf{W}^2}^2 \\
&\stackrel{(23e)}{=} (1-\beta)\left(\boldsymbol{g}(\mathbf{w}^t) + \langle \nabla \boldsymbol{g}(\mathbf{w}^t), \boldsymbol{\theta}^t - \mathbf{w}^t \rangle\right) + \beta\left(\boldsymbol{g}(\mathbf{w}^t) + \langle \nabla \boldsymbol{g}(\mathbf{w}^t), \mathbf{v}^{t+1} - \mathbf{w}^t \rangle\right) \\
&\quad + \frac{L}{m}\|\boldsymbol{\theta}^{t+1} - \mathbf{w}^t\|_{\mathbf{W}^2}^2 + \frac{\beta^2 \alpha}{2m}\|\boldsymbol{\theta}^{t+1} - \mathbf{w}^t\|_{\mathbf{I}-\mathbf{W}^2}^2 + \varepsilon_u(\boldsymbol{\theta}^{t+1}, \mathbf{w}^t) \\
&\leq (1-\beta)\boldsymbol{g}(\boldsymbol{\theta}^t) + \beta\left(\boldsymbol{g}(\mathbf{w}^t) + \langle \nabla \boldsymbol{g}(\mathbf{w}^t), \mathbf{v}^{t+1} - \mathbf{w}^t \rangle\right) \\
&\quad + \frac{L}{m}\|\boldsymbol{\theta}^{t+1} - \mathbf{w}^t\|_{\mathbf{W}^2}^2 + \frac{\beta^2 \alpha}{2m}\|\boldsymbol{\theta}^{t+1} - \mathbf{w}^t\|_{\mathbf{I}-\mathbf{W}^2}^2 + \varepsilon_u(\boldsymbol{\theta}^{t+1}, \mathbf{w}^t).
\end{aligned} \tag{66}
$$

Using (25a) while subtracting $-\boldsymbol{g}(\widehat{\boldsymbol{\theta}})$ on both sides yields

$$
\begin{aligned}
\boldsymbol{g}(\boldsymbol{\theta}^{t+1}) - \boldsymbol{g}(\widehat{\boldsymbol{\theta}}) &\leq (1-\beta)(\boldsymbol{g}(\boldsymbol{\theta}^t) - \boldsymbol{g}(\widehat{\boldsymbol{\theta}})) + \beta\langle \nabla \boldsymbol{g}(\mathbf{w}^t), \mathbf{v}^{t+1} - \widehat{\boldsymbol{\theta}} \rangle \\
&\quad - \frac{\beta\mu}{4m}\|\mathbf{w}^t - \widehat{\boldsymbol{\theta}}\|_{\mathbf{W}^2}^2 - \frac{\beta^3 \alpha}{2m}\|\mathbf{w}^t\|_{\mathbf{I}-\mathbf{W}^2}^2 + \beta\varepsilon_\ell(\widehat{\boldsymbol{\theta}}, \mathbf{w}^t) \\
&\quad + \frac{L}{m}\|\boldsymbol{\theta}^{t+1} - \mathbf{w}^t\|_{\mathbf{W}^2}^2 + \frac{\beta^2 \alpha}{2m}\|\boldsymbol{\theta}^{t+1} - \mathbf{w}^t\|_{\mathbf{I}-\mathbf{W}^2}^2 + \varepsilon_u(\boldsymbol{\theta}^{t+1}, \mathbf{w}^t).
\end{aligned} \tag{67}
$$

Invoking the optimality of $\mathbf{v}^{t+1}$ in (23d), we have

$$
\begin{aligned}
&\langle \nabla \boldsymbol{g}(\mathbf{w}^t), \mathbf{v}^{t+1} - \widehat{\boldsymbol{\theta}} \rangle \\
&\leq \left\langle \mathbf{y}^t - \nabla \boldsymbol{g}(\mathbf{w}^t) + \frac{\beta\alpha}{m}\left(\mathbf{v}^{t+1} - (1-\beta)\mathbf{v}^t - \beta\mathbf{z}^t\right), \widehat{\boldsymbol{\theta}} - \mathbf{v}^{t+1} \right\rangle \\
&= \left\langle \mathbf{y}^t + \frac{\beta^2\alpha}{m}(\mathbf{I}-\mathbf{W})\mathbf{w}^t - \nabla \boldsymbol{g}(\mathbf{w}^t) + \frac{\alpha\beta}{m}\left(\mathbf{v}^{t+1} - (1-\beta)\mathbf{v}^t - \beta\mathbf{w}^t\right), \widehat{\boldsymbol{\theta}} - \mathbf{v}^{t+1} \right\rangle \\
&\stackrel{(28)}{=} \left\langle \boldsymbol{\varepsilon}^t + \frac{\alpha\beta}{m}\left(\mathbf{v}^{t+1} - (1-\beta)\mathbf{v}^t - \beta\mathbf{w}^t\right), \widehat{\boldsymbol{\theta}} - \mathbf{v}^{t+1} \right\rangle.
\end{aligned} \tag{68}
$$

Using (68) in (67) we obtain

$$
\begin{aligned}
\boldsymbol{g}(\boldsymbol{\theta}^{t+1}) - \boldsymbol{g}(\widehat{\boldsymbol{\theta}}) &\leq (1-\beta)\left(\boldsymbol{g}(\boldsymbol{\theta}^t) - \boldsymbol{g}(\widehat{\boldsymbol{\theta}})\right) + \frac{\beta^2\alpha}{m}\langle \mathbf{v}^{t+1} - (1-\beta)\mathbf{v}^t - \beta\mathbf{w}^t, \widehat{\boldsymbol{\theta}} - \mathbf{v}^{t+1} \rangle \\
&\quad + \beta\left\langle \boldsymbol{\varepsilon}^t, \widehat{\boldsymbol{\theta}} - \mathbf{v}^{t+1} \right\rangle - \frac{\beta\mu}{4m}\|\mathbf{w}^t - \widehat{\boldsymbol{\theta}}\|_{\mathbf{W}^2}^2 - \frac{\beta^3\alpha}{2m}\|\mathbf{w}^t\|_{\mathbf{I}-\mathbf{W}^2}^2 + \beta\varepsilon_\ell(\widehat{\boldsymbol{\theta}}, \mathbf{w}^t) \\
&\quad + \frac{L}{m}\|\boldsymbol{\theta}^{t+1} - \mathbf{w}^t\|_{\mathbf{W}^2}^2 + \frac{\beta^2\alpha}{2m}\|\boldsymbol{\theta}^{t+1} - \mathbf{w}^t\|_{\mathbf{I}-\mathbf{W}^2}^2 + \varepsilon_u(\boldsymbol{\theta}^{t+1}, \mathbf{w}^t).
\end{aligned} \tag{69}
$$

We use now

$$
\beta\left(\mathbf{v}^{t+1} - (1-\beta)\mathbf{v}^t - \beta\mathbf{w}^t\right) \stackrel{(23e)}{=} \boldsymbol{\theta}^{t+1} - \mathbf{w}^t \tag{70}
$$

to bound the term

$$
\begin{aligned}
&\frac{\beta^2\alpha}{m}\langle \mathbf{v}^{t+1} - (1-\beta)\mathbf{v}^t - \beta\mathbf{w}^t, \widehat{\boldsymbol{\theta}} - \mathbf{v}^{t+1} \rangle \\
&= \frac{\beta^2\alpha}{2m}\|\widehat{\boldsymbol{\theta}} - (1-\beta)\mathbf{v}^t - \beta\mathbf{w}^t\|^2 - \frac{\beta^2\alpha}{2m}\|\mathbf{v}^{t+1} - (1-\beta)\mathbf{v}^t - \beta\mathbf{w}^t\|^2 - \frac{\beta^2\alpha}{2m}\|\widehat{\boldsymbol{\theta}} - \mathbf{v}^{t+1}\|^2 \\
&\stackrel{(70)}{=} \frac{\beta^2\alpha}{2m}\|\widehat{\boldsymbol{\theta}} - (1-\beta)\mathbf{v}^t - \beta\mathbf{w}^t\|^2 - \frac{\alpha}{2m}\|\boldsymbol{\theta}^{t+1} - \mathbf{w}^t\|^2 - \frac{\beta^2\alpha}{2m}\|\widehat{\boldsymbol{\theta}} - \mathbf{v}^{t+1}\|^2 \\
&\leq \frac{\beta^3\alpha}{2m}\|\mathbf{w}^t - \widehat{\boldsymbol{\theta}}\|^2 + \frac{(1-\beta)\beta^2\alpha}{2m}\|\widehat{\boldsymbol{\theta}} - \mathbf{v}^t\|^2 - \frac{\alpha}{2m}\|\boldsymbol{\theta}^{t+1} - \mathbf{w}^t\|^2 - \frac{\beta^2\alpha}{2m}\|\widehat{\boldsymbol{\theta}} - \mathbf{v}^{t+1}\|^2.
\end{aligned} \tag{71}
$$

Combining (69) and (71) yields

$$\boldsymbol{g}(\boldsymbol{\theta}^{t+1}) - \boldsymbol{g}(\widehat{\boldsymbol{\theta}}) \leq (1-\beta)\left(\boldsymbol{g}(\boldsymbol{\theta}^t) - \boldsymbol{g}(\widehat{\boldsymbol{\theta}})\right) - \frac{\beta}{2m}\left(\frac{\mu}{2} - \beta^2\alpha\right)\|\mathbf{w}^t - \widehat{\boldsymbol{\theta}}\|_{\mathbf{W}^2}^2$$
$$+ \frac{(1-\beta)\beta^2\alpha}{2m}\|\widehat{\boldsymbol{\theta}} - \mathbf{v}^t\|^2 - \frac{\beta^2\alpha}{2m}\|\widehat{\boldsymbol{\theta}} - \mathbf{v}^{t+1}\|^2 - \frac{\alpha}{2m}\|\boldsymbol{\theta}^{t+1} - \mathbf{w}^t\|^2$$
$$+ \frac{L}{m}\|\boldsymbol{\theta}^{t+1} - \mathbf{w}^t\|_{\mathbf{W}^2}^2 + \frac{\beta^2\alpha}{2m}\|\boldsymbol{\theta}^{t+1} - \mathbf{w}^t\|_{\mathbf{I}-\mathbf{W}^2}^2$$
$$+ \beta\langle\boldsymbol{\varepsilon}^t, \widehat{\boldsymbol{\theta}} - \mathbf{v}^{t+1}\rangle + \beta\varepsilon_\ell(\widehat{\boldsymbol{\theta}}, \mathbf{w}^t) + \varepsilon_u(\boldsymbol{\theta}^{t+1}, \mathbf{w}^t).$$

Using $\alpha\mathbf{I} \succeq 2L\mathbf{W}^2 + \beta^2\alpha(\mathbf{I} - \mathbf{W}^2)$, due to $\alpha \geq 2L$, rearranging terms, and using the definitions of $V^t$, we obtain the desired result as in (27). $\qquad\square$

### A.7.2 Proof of Lemma 2

Given $\varepsilon_\ell(\widehat{\boldsymbol{\theta}}, \mathbf{w}^t)$ as in (26), we notice that the argument points $z_i^t$ and $\sum_{j=1}^m w_{i,j}\theta_j^{t+1}$ are feasible for all $t$ and $i \in [m]$. We can then apply [1, Lemma 1] and conclude

$$\varepsilon_\ell(\widehat{\boldsymbol{\theta}}, \mathbf{w}^t) \leq \frac{1}{2m}\tau_\mu 8\Psi^2(\bar{\mathcal{M}})\|\mathbf{z}^t - \widehat{\boldsymbol{\theta}}\|^2 + \frac{\tau_\mu}{2}\nu^2.$$

Consider now $\varepsilon_u(\boldsymbol{\theta}^{t+1}, \mathbf{w}^t)$ (see (26)). Since

$$\mathbf{W}\boldsymbol{\theta}^{t+1} - \mathbf{z}^t \stackrel{(70)}{=} \beta\mathbf{W}\left(\mathbf{v}^{t+1} - (1-\beta)\mathbf{v}^t - \beta\mathbf{w}^t\right)$$

and $\mathcal{R}$ is a norm, it holds

$$\mathcal{R}^2\left(\sum_{j=1}^m w_{i,j}(\theta_j^{t+1} - z_i^t)\right) = \beta^2\mathcal{R}^2\left(\sum_{j=1}^m w_{i,j}\left(\beta v_j^{t+1} - (1-\beta)v_j^t - \beta w_j^t\right)\right)$$
$$\leq 3\beta^2\left(\mathcal{R}^2\left(\sum_{j=1}^m w_{i,j}v_j^{t+1} - \widehat{\theta}\right) + (1-\beta)^2\mathcal{R}^2\left(\sum_{j=1}^m w_{i,j}v_j^t - \widehat{\theta}\right) + \beta^2\mathcal{R}^2\left(z_j^t - \widehat{\theta}\right)\right).$$

Again, invoking feasibility of the $z, v$-iterates and [1, Lemma 1] we deduce (31). $\qquad\square$

### A.7.3 Proof of Proposition 2

By definition

$$\langle\boldsymbol{\varepsilon}^t, \widehat{\boldsymbol{\theta}} - \mathbf{v}^{t+1}\rangle = \underbrace{\langle\mathbf{y}^t - \mathbf{W}\nabla\mathbf{F}(\mathbf{W}\mathbf{w}^t), \widehat{\boldsymbol{\theta}} - \mathbf{v}^{t+1}\rangle}_{\texttt{term I}} + \underbrace{\frac{\beta^2\alpha}{m}\langle(\mathbf{W}^2 - \mathbf{W})\mathbf{w}^t, \widehat{\boldsymbol{\theta}} - \mathbf{v}^{t+1}\rangle}_{\texttt{term II}}. \qquad (72)$$

We bound the two terms separately.

For `term II`, we have

$$\texttt{term II} = \frac{\beta^2\alpha}{m}\langle(\mathbf{W} - \mathbf{I})(\mathbf{w}^t), (\mathbf{W} - \mathbf{J})(\widehat{\boldsymbol{\theta}} - \mathbf{v}^{t+1})\rangle$$
$$\stackrel{(a)}{\leq} \frac{\beta^2\alpha\rho}{2m}\|\mathbf{w}^t\|_{\mathbf{I}-\mathbf{W}}^2 + \frac{\beta^2\alpha\rho}{2m}\|\mathbf{v}^{t+1}\|_{\mathbf{I}-\mathbf{W}}^2$$
$$\stackrel{(23a)}{\leq} \frac{\beta^2\alpha\rho}{2m(1+\beta)}\|\boldsymbol{\theta}^t\|_{\mathbf{I}-\mathbf{W}}^2 + \frac{\beta^3\alpha\rho}{2m(1+\beta)}\|\mathbf{v}^t\|_{\mathbf{I}-\mathbf{W}}^2 + \frac{\beta^2\alpha\rho}{2m}\|\mathbf{v}^{t+1}\|_{\mathbf{I}-\mathbf{W}}^2$$

where in (a) we have used the fact that $(\mathbf{I} - \mathbf{W})^{1/2}$ and $(\mathbf{W} - \mathbf{J})^{1/2}$ commute.

We work now with `term I`. Due to the favorable behavior along the consensus direction, we bound `term I` considering separately the inner product along the consensus space and its orthogonal complement, that is,

$$\texttt{term I} = \underbrace{\langle\mathbf{y}^t - \mathbf{W}\nabla\boldsymbol{F}(\mathbf{W}\mathbf{w}^t), \mathbf{J}(\widehat{\boldsymbol{\theta}} - \mathbf{v}^{t+1})\rangle}_{\texttt{term I(a)}} + \underbrace{\langle\mathbf{y}^t - \mathbf{W}\nabla\boldsymbol{F}(\mathbf{W}\mathbf{w}^t), (\mathbf{I} - \mathbf{J})(\widehat{\boldsymbol{\theta}} - \mathbf{v}^{t+1})\rangle}_{\texttt{term I(b)}}.$$

We will leverage the following properties:

$$\mathbf{J}\mathbf{y}^t = \mathbf{J}\nabla \boldsymbol{f}(\mathbf{W}\mathbf{w}^t), \tag{73a}$$

$$\mathbf{J}\nabla \boldsymbol{f}(\mathbf{J}\mathbf{w}^t) = \mathbf{J}\nabla \boldsymbol{F}(\mathbf{J}\mathbf{w}^t). \tag{73b}$$

where in (73a) we explicitly used the initialization $\mathbf{y}^{-1} = \nabla \boldsymbol{f}(\mathbf{W}\mathbf{w}^{-1})$, and (73b) follows by inspection.

We proceed working with `term I(a)`, as follows:

$$\begin{aligned}
\texttt{term I(a)} \overset{(73a)}{=} & \langle \nabla \boldsymbol{f}(\mathbf{W}\mathbf{w}^t) - \nabla \boldsymbol{F}(\mathbf{W}\mathbf{w}^t), \mathbf{J}(\widehat{\boldsymbol{\theta}} - \mathbf{v}^{t+1}) \rangle \\
\overset{(73b)}{=} & \langle \nabla \boldsymbol{f}(\mathbf{W}\mathbf{w}^t) - \nabla \boldsymbol{f}(\mathbf{J}\mathbf{w}^t) + \nabla \boldsymbol{F}(\mathbf{J}\mathbf{w}^t) - \nabla \boldsymbol{F}(\mathbf{W}\mathbf{w}^t), \mathbf{J}(\widehat{\boldsymbol{\theta}} - \mathbf{v}^{t+1}) \rangle \\
\overset{(a)}{=} & \langle (\nabla^2 \boldsymbol{f})(\mathbf{W} - \mathbf{J})(\mathbf{w}^t - \widehat{\boldsymbol{\theta}}), \mathbf{J}(\widehat{\boldsymbol{\theta}} - \mathbf{v}^{t+1}) \rangle,
\end{aligned}$$

where in (a) we used the following two facts: the functions are quadratic, with $\nabla^2 \boldsymbol{f}$ denoting the Hessian of the quadratic function $\boldsymbol{f}$; and $(\mathbf{I} - \mathbf{J})(\nabla^2 \boldsymbol{F})\mathbf{J} = \mathbf{0}$.

We move to `term I(b)`. Using

$$\mathbf{y}^t = \mathbf{W}\nabla \boldsymbol{f}(\mathbf{W}\mathbf{w}^t) + \sum_{i=0}^{t-1} (\mathbf{W} - \mathbf{I})\mathbf{W}^{t-i}\nabla \boldsymbol{f}(\mathbf{W}\mathbf{w}^i)$$

we have:

$$\begin{aligned}
\texttt{term I(b)} = & -\langle (\mathbf{W} - \mathbf{J})^{t+1}\nabla \boldsymbol{f}(\widehat{\boldsymbol{\theta}}), \widehat{\boldsymbol{\theta}} - \mathbf{v}^{t+1} \rangle \\
& + \langle (\mathbf{W} - \mathbf{J})(\nabla \boldsymbol{f}(\mathbf{W}\mathbf{w}^t) - \nabla \boldsymbol{f}(\widehat{\boldsymbol{\theta}}) - \nabla \boldsymbol{F}(\mathbf{W}\mathbf{w}^t)), \widehat{\boldsymbol{\theta}} - \mathbf{v}^{t+1} \rangle \\
& + \sum_{i=0}^{t-1} \langle (\mathbf{W} - \mathbf{J})^{t-i} \left( \mathbf{W}(\nabla \boldsymbol{f}(\mathbf{W}\mathbf{w}^i) - \nabla \boldsymbol{f}(\widehat{\boldsymbol{\theta}})) + \nabla \boldsymbol{f}(\widehat{\boldsymbol{\theta}}) - \nabla \boldsymbol{f}(\mathbf{W}\mathbf{w}^i) \right), \widehat{\boldsymbol{\theta}} - \mathbf{v}^{t+1} \rangle \\
\overset{(a)}{=} & -\langle (\mathbf{W} - \mathbf{J})^{t+1}\nabla \boldsymbol{f}(\widehat{\boldsymbol{\theta}}), \widehat{\boldsymbol{\theta}} - \mathbf{v}^{t+1} \rangle \\
& + \langle (\nabla^2 \boldsymbol{f})(\mathbf{z}^t - \widehat{\boldsymbol{\theta}}), (\mathbf{W} - \mathbf{J})(\widehat{\boldsymbol{\theta}} - \mathbf{v}) \rangle + \langle (\nabla^2 \boldsymbol{F})(\mathbf{z}^t - \widehat{\boldsymbol{\theta}}), (\mathbf{W} - \mathbf{J})(\mathbf{v}^{t+1} - \widehat{\boldsymbol{\theta}}) \rangle \\
& + \sum_{i=0}^{t-1} \langle (\mathbf{W} - \mathbf{J})^{t-i}(\mathbf{W} - \mathbf{I})(\nabla^2 \boldsymbol{f})(\mathbf{z}^i - \widehat{\boldsymbol{\theta}}), \widehat{\boldsymbol{\theta}} - \mathbf{v}^{t+1} \rangle.
\end{aligned}$$

where in (a) we used the fact that the functions are quadratic and that $\mathbf{J}\nabla \boldsymbol{F}(\widehat{\boldsymbol{\theta}}) = \nabla \boldsymbol{F}(\widehat{\boldsymbol{\theta}})$).

Using the above expressions of `term I(a)` and `term I(b)`, `term I` reads

$$\begin{aligned}
\texttt{term I} = & \langle (\nabla^2 \boldsymbol{f})(\mathbf{W} - \mathbf{J})(\mathbf{w}^t - \widehat{\boldsymbol{\theta}}), \mathbf{J}(\widehat{\boldsymbol{\theta}} - \mathbf{v}^{t+1}) \rangle - \langle (\mathbf{W} - \mathbf{J})^{t+1}\nabla \boldsymbol{f}(\widehat{\boldsymbol{\theta}}), \widehat{\boldsymbol{\theta}} - \mathbf{v}^{t+1} \rangle \\
& + \langle (\nabla^2 \boldsymbol{f})(\mathbf{z}^t - \widehat{\boldsymbol{\theta}}), (\mathbf{W} - \mathbf{J})(\widehat{\boldsymbol{\theta}} - \mathbf{v}^{t+1}) \rangle + \langle (\nabla^2 \boldsymbol{F})(\mathbf{z}^t - \widehat{\boldsymbol{\theta}}), (\mathbf{W} - \mathbf{J})(\mathbf{v}^{t+1} - \widehat{\boldsymbol{\theta}}) \rangle \\
& + \sum_{i=0}^{t-1} \langle (\mathbf{W} - \mathbf{J})^{t-i}(\mathbf{W} - \mathbf{I})(\nabla^2 \boldsymbol{f})(\mathbf{z}^i - \widehat{\boldsymbol{\theta}}), \widehat{\boldsymbol{\theta}} - \mathbf{v}^{t+1} \rangle.
\end{aligned}$$

To properly control `term I`, we need to bounds terms therein having the following structure:

$$|\langle (\nabla^2 \boldsymbol{f})(\mathbf{w} - \widehat{\boldsymbol{\theta}}), (\mathbf{W} - \mathbf{J})^k(\widehat{\boldsymbol{\theta}} - \mathbf{v}) \rangle| \quad \text{and} \quad |\langle (\nabla^2 \boldsymbol{f})(\mathbf{J})(\mathbf{w} - \widehat{\boldsymbol{\theta}}), (\mathbf{W} - \mathbf{J})^k(\widehat{\boldsymbol{\theta}} - \mathbf{v}) \rangle|,$$

$$|\langle (\nabla^2 \boldsymbol{F})(\mathbf{w} - \widehat{\boldsymbol{\theta}}), (\mathbf{W} - \mathbf{J})(\mathbf{v} - \widehat{\boldsymbol{\theta}}) \rangle|,$$

$$\langle (\mathbf{W} - \mathbf{J})^k \nabla \boldsymbol{f}(\widehat{\boldsymbol{\theta}}), \widehat{\boldsymbol{\theta}} - \mathbf{v} \rangle,$$

for $\mathbf{w}$ and $\mathbf{v}$ with feasible $d$-blocks, and integer $k$.

The following lemma provides suitable bounds for these terms.

**Lemma 3.** *For any* $\mathbf{w} = [w_1^\top, \ldots, w_m^\top]^\top$ *and* $\mathbf{v} = [v_1^\top, \ldots, v_m^\top]^\top$, *with* $w_i, v_i$'s *feasible for the ERM* (3), *the following hold:*

$$
|\langle (\nabla^2 \boldsymbol{f})(\mathbf{w} - \widehat{\boldsymbol{\theta}}), (\mathbf{W} - \mathbf{J})^k (\widehat{\boldsymbol{\theta}} - \mathbf{v}) \rangle|
$$
$$
\leq \frac{3m\sqrt{m}\rho^k(2\gamma_\ell + 8\Psi^2(\bar{\mathcal{M}})\tau_\ell)}{2m} \left( \|\mathbf{w} - \widehat{\boldsymbol{\theta}}\|^2 + \|\mathbf{v} - \widehat{\boldsymbol{\theta}}\|^2 \right) + \frac{m^2\sqrt{m}\rho^k\tau_\ell}{2m} 18\nu^2, \quad (74)
$$
$$
|\langle (\nabla^2 \boldsymbol{F})(\mathbf{w} - \widehat{\boldsymbol{\theta}}), (\mathbf{W} - \mathbf{J}))\widehat{\boldsymbol{\theta}} - \mathbf{v}) \rangle|
$$
$$
\leq \frac{3m^2\sqrt{m}\rho \left( 2L + 8\tau_L\Psi^2(\bar{\mathcal{M}}) \right)}{2m} \left( \|\mathbf{w} - \widehat{\boldsymbol{\theta}}\|^2 + \|\mathbf{v} - \widehat{\boldsymbol{\theta}}\|^2 \right) + \frac{m^2\sqrt{m}\rho\tau_L}{2m} 18\nu^2, \quad (75)
$$

*and*

$$
\langle (\mathbf{W} - \mathbf{J})^k \nabla \boldsymbol{f}(\widehat{\boldsymbol{\theta}}), \widehat{\boldsymbol{\theta}} - \mathbf{v} \rangle \tag{76}
$$
$$
\leq 2R\sqrt{m}m\rho^k \frac{\sum_{i=1}^m \mathcal{R}^*(\nabla f_i(\theta^\star))}{m} + \frac{3m\sqrt{m}\rho^k(2\gamma_\ell + 8\Psi^2(\bar{\mathcal{M}})\tau_\ell)}{2m} \left( \|\mathbf{v} - \widehat{\boldsymbol{\theta}}\|^2 + \|\widehat{\boldsymbol{\theta}} - \theta^\star\|^2 \right)
$$
$$
+ \frac{m^2\sqrt{m}\rho^k\tau_\ell}{2m} 18\nu^2. \tag{77}
$$

*Proof.* See Appendix A.7.4. ◻

Using (74), we have

$$
\langle (\nabla^2 \boldsymbol{f})(\mathbf{W} - \mathbf{J})(\mathbf{w}^t - \widehat{\boldsymbol{\theta}}), \mathbf{J}(\widehat{\boldsymbol{\theta}} - \mathbf{v}^{t+1}) \rangle
$$
$$
\leq \frac{3m\sqrt{m}\rho(2\gamma_\ell + 3\Psi^2(\bar{\mathcal{M}})\tau_\ell)}{2m} \left( \|\mathbf{w}^t - \widehat{\boldsymbol{\theta}}\|^2 + \|\mathbf{v}^{t+1} - \widehat{\boldsymbol{\theta}}\|^2 \right) + \frac{m\sqrt{m}\rho\tau_\ell}{2m} 18\nu^2
$$
$$
\leq \frac{3m^2\sqrt{m}\rho(2\gamma_\ell + 3\Psi^2(\bar{\mathcal{M}})\tau_\ell)}{2m} \left( \frac{1}{1+\beta}\|\boldsymbol{\theta}^t\|_{\mathbf{I}-\mathbf{W}^2}^2 + \frac{\beta}{1+\beta}\|\mathbf{v}^t - \widehat{\boldsymbol{\theta}}\|^2 + \|\mathbf{v}^{t+1} - \widehat{\boldsymbol{\theta}}\|^2 \right) + \frac{m^2\sqrt{m}\rho\tau_\ell}{2m} 18\nu^2,
$$

where in the last inequality we used

$$
\|\mathbf{w}^t - \widehat{\boldsymbol{\theta}}\|^2 = \|\mathbf{z}^t - \widehat{\boldsymbol{\theta}}\|^2 + \|\mathbf{w}^t\|_{\mathbf{I}-\mathbf{W}^2}^2 \leq \|\mathbf{z}^t - \widehat{\boldsymbol{\theta}}\|^2 + \frac{1}{1+\beta}\|\boldsymbol{\theta}^t\|_{\mathbf{I}-\mathbf{W}^2}^2 + \frac{\beta}{1+\beta}\|\mathbf{v}^t\|_{\mathbf{I}-\mathbf{W}^2}^2.
$$

Using similar path to bound the other terms in `term I` and combining together the final bounds of `term I` and `term II`, we obtain

$$
\langle \varepsilon^t, \widehat{\boldsymbol{\theta}} - \mathbf{v}^{t+1} \rangle \leq \frac{3m\sqrt{m}\rho(2\gamma_\ell + 8\Psi^2(\bar{\mathcal{M}})\tau_\ell)}{2m} \left( \frac{1}{1+\beta}\|\boldsymbol{\theta}^t\|_{\mathbf{I}-\mathbf{W}^2}^2 + \frac{\beta}{1+\beta}\|\mathbf{v}^t - \widehat{\boldsymbol{\theta}}\|^2 + \|\mathbf{v}^{t+1} - \widehat{\boldsymbol{\theta}}\|^2 \right)
$$
$$
+ \frac{m^2\sqrt{m}\rho\tau_\ell}{2m} 18\nu^2 + 2R\sqrt{m}m\rho^{t+1} \sum_{i=1}^m \frac{\mathcal{R}^*(\nabla f_i(\theta^\star))}{m}
$$
$$
+ \frac{3m\sqrt{m}\rho^{t+1}(2\gamma_\ell + 8\Psi^2(\bar{\mathcal{M}})\tau_\ell)}{2m} \left( \|\widehat{\boldsymbol{\theta}} - \theta^\star\|^2 + \|\mathbf{v}^{t+1} - \widehat{\boldsymbol{\theta}}\|^2 \right) + \frac{m^2\sqrt{m}\rho^{t+1}\tau_\ell}{2m} 18\nu^2
$$
$$
+ \frac{3m\sqrt{m}\rho(2\gamma_\ell + 8\Psi^2(\bar{\mathcal{M}})\tau_\ell)}{2m} \left( \|\mathbf{z}^t - \widehat{\boldsymbol{\theta}}\|^2 + \|\mathbf{v}^{t+1} - \widehat{\boldsymbol{\theta}}\|^2 \right) + \frac{m^2\sqrt{m}\rho\tau_\ell}{2m} 18\nu^2
$$
$$
+ \frac{3m\sqrt{m}\rho(2L + 8\tau_L\Psi^2(\bar{\mathcal{M}}))}{2m} \left( \|\widehat{\boldsymbol{\theta}} - \mathbf{v}^{t+1}\|^2 + \|\mathbf{z}^t - \widehat{\boldsymbol{\theta}}\|^2 \right) + \frac{m^2\sqrt{m}\rho\tau_L}{2m} 18\nu^2
$$
$$
+ \sum_{i=0}^{t-1} (1+\rho)\rho^{t-i} \left( \frac{3m\sqrt{m}(2\gamma_\ell + 8\Psi^2(\bar{\mathcal{M}})\tau_\ell)}{2m} \left( \|\mathbf{z}^i - \widehat{\boldsymbol{\theta}}\|^2 + \|\mathbf{v}^{t+1} - \widehat{\boldsymbol{\theta}}\|^2 \right) + \frac{m^2\sqrt{m}\tau_\ell}{2m} 18\nu^2 \right)
$$
$$
+ \frac{\beta^2\alpha\rho}{2m(1+\beta)}\|\boldsymbol{\theta}^t\|_{\mathbf{I}-\mathbf{W}^2}^2 + \frac{\beta^3\alpha\rho}{2m(1+\beta)}\|\mathbf{v}^t - \widehat{\boldsymbol{\theta}}\|^2 + \frac{\beta^2\alpha\rho}{2m}\|\mathbf{v}^{t+1} - \widehat{\boldsymbol{\theta}}\|^2.
$$

By properly grouping terms, we obtain the desired result

$$\langle \varepsilon^t, \widehat{\boldsymbol{\theta}} - \mathbf{v}^{t+1} \rangle \leq 2R\sqrt{m}m\rho^{t+1} \sum_{i=1}^{m} \frac{\mathcal{R}^*(\nabla f_i(\theta^\star))}{m} + \frac{3m\sqrt{m}\rho^{t+1}(2\gamma_\ell + 8\Psi^2(\bar{\mathcal{M}})\tau_\ell)}{2m} \|\widehat{\boldsymbol{\theta}} - \boldsymbol{\theta}^\star\|^2$$

$$+ \left( \frac{\beta^2\alpha\rho}{2m} + \frac{3m\sqrt{m}\rho}{2m} \left( \left(3 + \frac{2\rho}{1-\rho}\right)(2\gamma_\ell + 8\Psi^2(\bar{\mathcal{M}})\tau_\ell) + (2L + 8\tau_L\Psi^2(\bar{\mathcal{M}})) \right) \right) \|\mathbf{v}^{t+1} - \widehat{\boldsymbol{\theta}}\|^2$$

$$+ \frac{3m\sqrt{m}\rho}{2m} \left( 2(\gamma_\ell + L) + 8\Psi^2(\bar{\mathcal{M}})(\tau_\ell + \tau_L) \right) \|\mathbf{z}^t - \widehat{\boldsymbol{\theta}}\|^2$$

$$+ \frac{3m\sqrt{m}(2\gamma_\ell + 8\Psi^2(\bar{\mathcal{M}})\tau_\ell)}{2m} \sum_{i=0}^{t-1} (1+\rho)\rho^{t-i}\|\mathbf{z}^i - \widehat{\boldsymbol{\theta}}\|^2$$

$$\left( \frac{\beta^2\alpha\rho}{2m} + \frac{3m\sqrt{m}\rho}{2m} (2\gamma_\ell + 8\Psi^2(\bar{\mathcal{M}})\tau_\ell) \right) \left( \frac{1}{1+\beta} \|\boldsymbol{\theta}^t\|_{\mathbf{I}-\mathbf{W}^2}^2 + \frac{\beta}{1+\beta} \|\mathbf{v}^t - \widehat{\boldsymbol{\theta}}\|^2 \right)$$

$$+ \frac{m^2\sqrt{m}\rho}{2m} 18\Psi^2(\bar{\mathcal{M}})\nu^2 \left( \left(3 + \frac{2\rho}{1-\rho}\right)\tau_\ell + \tau_L \right).$$

$\square$

### A.7.4 Proof of Lemma 3

For convenience, denote by $b_{i,j}^k$ the $i, j$ element of the matrix $(W - J)^k$. Then, it holds

$$|\langle(\nabla^2\boldsymbol{f})(\mathbf{w} - \widehat{\boldsymbol{\theta}}), (\mathbf{W} - \mathbf{J})^k(\widehat{\boldsymbol{\theta}} - \mathbf{v})\rangle| = \frac{1}{m} \left| \sum_{i=1}^{m} \left\langle (\nabla^2 f_i)(w_i - \widehat{\theta}), \sum_{j=1}^{m} b_{i,j}^k(\widehat{\theta} - v_j) \right\rangle \right|$$

$$\leq \frac{1}{m} \sum_{i=1}^{m} \sum_{j=1}^{m} |b_{i,j}^k| \left| \left\langle (\nabla^2 f_i)(w_i - \widehat{\theta}), \widehat{\theta} - v_j \right\rangle \right|.$$

Then, under Assumption 2 it follows that

$$\left| \left\langle (\nabla^2 f_i)(w_i - \widehat{\theta}), \widehat{\theta} - v_i \right\rangle \right| \leq \frac{1}{2} \left| \langle(\nabla^2 f_i)(w_i - \widehat{\theta}), w_i - \widehat{\theta}\rangle \right| + \frac{1}{2} \left| \left\langle (\nabla^2 f_i)(v_j - \widehat{\theta}), v_k - \widehat{\theta} \right\rangle \right|$$

$$+ \frac{1}{2} \left| \langle(\nabla^2 f_i)(w_j - v_i), w_j - v_i\rangle \right|$$

$$\leq \gamma_\ell \left( \|w_i - \widehat{\theta}\|^2 + \|v_j - \widehat{\theta}\|^2 + \|w_i - v_j\|^2 \right)$$

$$+ \frac{\tau_\ell}{2} \left( \mathcal{R}^2(w_i - \widehat{\theta}) + \mathcal{R}^2(v_j - \widehat{\theta}) + \mathcal{R}^2(v_j - w_i) \right).$$

Under Assumption 5, it holds $|b_{i,j}| \leq \sqrt{m}\rho^k$ [26], yielding

$$|\langle(\nabla^2\boldsymbol{f})(\mathbf{w} - \widehat{\boldsymbol{\theta}}), (\mathbf{W} - \mathbf{J})^k(\widehat{\boldsymbol{\theta}} - \mathbf{v})\rangle|$$

$$\leq \sum_{i=1}^{m} \sum_{j=1}^{m} \frac{\sqrt{m}\rho^k 6\gamma_\ell}{2m} \left( \|w_i - \widehat{\theta}\|^2 + \|v_j - \widehat{\theta}\|^2 \right) + \sum_{i=1}^{m} \sum_{j=1}^{m} \frac{\sqrt{m}\rho^k\tau_\ell}{2m} \left( 3\mathcal{R}^2(w_i - \widehat{\theta}) + 3\mathcal{R}^2(v_j - \widehat{\theta}) \right).$$

Because $w_i$ and $v_j$ are feasible for all $j, i \in [m]$, we can use in [1, Lemma 1] and obtain

$$|\langle(\nabla^2\boldsymbol{f})(\mathbf{w} - \widehat{\boldsymbol{\theta}}), (\mathbf{W} - \mathbf{J})^k(\widehat{\boldsymbol{\theta}} - \mathbf{v})\rangle|$$

$$\leq \sum_{i=1}^{m} \sum_{j=1}^{m} \frac{6\sqrt{m}\rho^k\gamma_\ell}{2m} \left( \|w_i - \widehat{\theta}\|^2 + \|v_j - \widehat{\theta}\|^2 \right)$$

$$+ \sum_{i=1}^{m} \sum_{j=1}^{m} \frac{3\sqrt{m}\rho^k\tau_\ell}{2m} \left( 8\Psi^2(\bar{\mathcal{M}})(\|w_i - \widehat{\theta}\|^2 + \|v_j - \widehat{\theta}\|^2) + 6\nu^2 \right)$$

$$= \frac{3m\sqrt{m}\rho^k(2\gamma_\ell + 8\Psi^2(\bar{\mathcal{M}})\tau_\ell)}{2m} \left( \|\mathbf{w} - \widehat{\boldsymbol{\theta}}\|^2 + \|\mathbf{v} - \widehat{\boldsymbol{\theta}}\|^2 \right) + \frac{\sqrt{m}m^2\rho^k\tau_\ell}{2m} 18\nu^2,$$

where $\nu^2$ is defined in (29).

The proof of (75) follows identical steps as above, with the only difference of using the RSM property in Assumption 1.

We prove now (77). Denote by $b_{i,j}^k$ the $i, j$ element of the matrix $(W - J)^k$. We have

$$\langle (\mathbf{W} - \mathbf{J})^k \nabla \boldsymbol{f}(\widehat{\boldsymbol{\theta}}), \widehat{\boldsymbol{\theta}} - \mathbf{v} \rangle = \langle (\mathbf{W} - \mathbf{J})^k \nabla \boldsymbol{f}(\boldsymbol{\theta}^\star), \widehat{\boldsymbol{\theta}} - \mathbf{v} \rangle + \langle (\mathbf{W} - \mathbf{J})^k (\nabla^2 \boldsymbol{f})(\widehat{\boldsymbol{\theta}} - \boldsymbol{\theta}^\star), \widehat{\boldsymbol{\theta}} - \mathbf{v} \rangle$$

$$= \frac{1}{m} \sum_{i=1}^m \sum_{j=1}^m b_{i,j}^k \langle \nabla f_j(\boldsymbol{\theta}^\star), \widehat{\boldsymbol{\theta}} - v_i \rangle + \langle (\mathbf{W} - \mathbf{J})^k (\nabla^2 \boldsymbol{f})(\widehat{\boldsymbol{\theta}} - \boldsymbol{\theta}^\star), \widehat{\boldsymbol{\theta}} - \mathbf{v} \rangle. \tag{78}$$

We now proceed to upper bound the terms on the RHS. Under Assumption 5 it can be shown that $|b_{i,j}^k| \leq \sqrt{m}\rho^k$ [26], therefore using Hölder's inequality along with feasibility of each $v_i$ and $\widehat{\theta}$, we obtain

$$\frac{1}{m} \sum_{i=1}^m \sum_{j=1}^m b_{i,j}^k \langle \nabla f_j(\boldsymbol{\theta}^\star), \widehat{\theta} - v_i \rangle \leq 2R\sqrt{m}m\rho^k \frac{\sum_{i=1}^m \mathcal{R}^*(\nabla f_i(\theta^\star))}{m}.$$

For the second term in (78) we can use the upper bound in (74). □