# OpenReview forum: "Acceleration in Distributed Sparse Regression"
_NeurIPS.cc/2022/Conference — NeurIPS 2022 Accept_

### Official Review · Reviewer_2ihf · 2022-07-10

**Rating:** 7
**Confidence:** 3
**Soundness:** 3 good
**Presentation:** 3 good
**Contribution:** 3 good

**Summary:**

The paper proposes an accelerated algorithm for distributed sparse regression in high dimensions.
They combine accelerated Nesterov’s proximal gradient with consensus and gradient tracking mechanisms.
They show it converges globally at a linear rate, achieving both optimal iteration complexity and communication complexity under the first-order oracle.

**Questions:**

Questions:

1. The communication complexity is nearly optimal up to log factors. Is it possible to save the $\log m$ dependence in (5)?

2. Can you upper bound the $\Delta^2$ defined in (10) in terms of interest parameters like $m, N, d$？

3. Theorem 2 requires $\rho$ to be sufficiently small (see (16)). Though the author says we can run multiple rounds of communications to ensure (16) holds, some previous work doesn’t require it (e.g., [11]). Could I say this is because of the used accelerated technique, the resulting analysis needs to carefully bound residual errors?

4. In the proof, the author makes use of a Lyapunov-like function to derive convergence. See (23). My question is why you use $\hat{\theta}$ in the definition of $(23)$ not $\theta^*$?

5. An extension question: the paper studies the case where all samples are generated according to $\mathbb{P}$, while in some applications like Federated Learning (FL), the data points on different devices will conform to different distributions. Do you think your proposed method will still work in the data-heterogeneous setting? If not, how to overcome it?

=============Post Rebuttal=============

I have read the author's responses. All of my questions have been addressed. I keep my score.

**Limitations:**

See the questions.

**Strengths And Weaknesses:**

Originality: The problem setting is quite different from many previous results. The paper considers the high-dimension setting where the ambient dimension $d$ could be much larger than the number of samples $N$, failing many previous algorithms. Though the proposed solution is a combination of two popular techniques, the analysis is new. The analysis setting follows from [1] but has been modified to adapt to the distributed setting.

Quality: The paper is technically sound with all theoretical claims supported. The experiment setting is clearly described.

Clarity: The paper is overall well-written and easy to follow.

Significance: The paper answers an important question in distributed learning and fills the blank of methods and theories in sparse distributed learning in high dimensions. It might be important.

---

> ### Author Response · Authors · 2022-08-02
> **Reply  to Reviewer 2ihf-Part I**
>
>
>  We than the Reviewer for the positive and careful assessment of the paper and insightful comments. Our reply to her/his questions follows.
>
> $\textbf{1.}$ The Reviewer is right:  the communication complexity is optimal only up to log-factors.   In particular,  such a log-term is a consequence of the condition on the network connectivity $\rho$, see Eq. (16) or (18) in the paper, which guarantees that centralized statistical consistency is achieved over network, {\it without requiring any condition on the local sample size} (agents can have as many as one sample!). If the network already satisfies such a condition on $\rho$ (roughly speaking, it is sufficient connected), then the number of communications to an $\varepsilon$-solution will match the total number of iterations $\mathcal{O}(\sqrt{\kappa}\log1\varepsilon)$, and will be optimal. However for graphs with arbitrary topology, this may not be  the case, and multiple rounds of communications per iterations are needed to enforce such a condition on $\rho$, resulting in the log-factor in the communication complexity. We remark that such a condition on $\rho$ is not an artifact of the proof but a structural requirement from the algorithm to achieve *centralized* statistical consistency. In fact,   in Figure 2 in the paper we showed that  keeping the connectivity constant, violating eventually (for large $m$) the aforementioned condition on $\rho$, breaks down  convergence to a centralized statistically optimal solution. We conjecture that the explicit use of  gradient tracking, coupled in the primal domain with consensus updates,  is  responsible for such a condition on $\rho$.  The design of new distributed and accelerated schemes in high-dimension overcoming this limitation remains an open problem. We added some conclusions to the paper (see item 2 in the section of ``General Comments'') where we highlighted this open problem.  Thanks again for the comment.
>
> $\textbf{2.}$  The expression of $\Delta^2$ in (10) depends on   $\mathcal{R}^2(\Delta^{\star})$ and  $\\|\Delta^{\star}\\|^2$, whose specific expressions (bounds) as function of $n,d,m, N$ depend on the statistical model under consideration.  For instance,    $\mathcal{R}(\Delta^{\star}) \leq 2 \Psi(\bar{\mathcal{M}})||\Delta^{\star}|| + 2 \mathcal{R}(\Pi_{\bar{\mathcal{M}}^{\perp}}(\theta^\star))$, and the requested bounds for $\Psi(\bar{\mathcal{M}})$ and $\mathcal{R}(\Pi_{\bar{\mathcal{M}}^{\perp}}(\theta^\star))$ were given in Eq. (63) in the supporting document for the gaussian linear regression model. For the same model,   $||\Delta^{\star}||^2$, which is   the   mean square error achieved by the empirical risk minimizer, reads $||\Delta^{\star}||^2 \leq \frac{16 \sigma^2 s \log (d)}{\mu^2 N}$ (See [29]) Similarly, for the other statistical models the aforementioned quantities are in  appendix A under lines 675, 677 and 695.
>
> $ \textbf{3.}$ This is a very good observation/question. In the distributed algorithms there is in general a dependence (tension) of the admissible  step-size values  on the network connectivity $\rho$.  Specifically, if one wants to choose the stepsize  independently on the network connectivity  (e.g., matching that of the centralized gradient), a condition limiting the value of the network connectivity is needed, to permit such "large'' step values. On the other hand, if one does not want to enforce any condition on $\rho$, then inevitably the stepsize value is limited by a quantity that depends (explicitly or implicitly) on the fact that $\rho$ is not constrained.  In our paper, we target centralized statistical consistency and stepsize values matching those of the centralized proximal gradient; as by product, we ended up with a condition on $\rho$. The specific scaling of $\rho$ is algorithmic dependent (some algorithms can have a more favorable scaling than others). For the proposed gradient-tracking based method, we obtain conditions for $\rho$ scaling inversely with (a polynomial of) the network size (rather than the ambient dimension $d$, which  is larger than $m$ in high-dimension).
>
> It is true that the work in [11] does not enforce any condition on the underlying network. However, consistently with the above argument,   [11] requires the step-size to be chosen as $\mathcal{O}(d^{-1})$, yielding  $\mathcal{O}(d \log (1/\varepsilon))$ iterations  to achieve $\varepsilon$ accuracy to statistical precision. Note that such a scaling with the ambient dimension is sensitive in high-dimension, because the typical regimes of interest are the asymptotic $d/N\to \infty$. This suggests that the stepsize should be chosen to be independent of the dimension. Note that a condition on $\rho$ limited by $d^{-1}$ would be less disruptive  on the total complexity, because it would lead to a total number of iterations scaling logarithmically with $d$ rather than linear (as in the case $\rho$ is free and the stepsize is chosen accordingly).

---

> > ### Author Response · Authors · 2022-08-02
> > **Reply to Reviewer 2ihf - Part II**
> >
> > $\textbf{4.}$ The analysis can be carried over with $\boldsymbol{\theta}^{\star}$ instead of $\hat{\boldsymbol{\theta}}.$ However, because the Lyapunov function is built using the empirical risk as opposed to the population risk $\hat{\boldsymbol{\theta}}$ becomes more convenient to use as it does not require to carry over a further additional error.
> >
> > $\textbf{5.}$ This is an interesting question. In the low dimension setting (as most of FL works), it is known that in fact gradient-tracking methods are good candidate to deal with data heterogeneity (under mild assumptions on local function dissimilarity). This would suggest that the proposed method has some potential to work in high dimension as well. However the tricky part will be understanding whether one can preserve centralized sample complexity at the cost of no extra (or restrictive) assumptions. We expect that, as in the low-dimensional case, some condition controlling the  discrepancy of the local distribution s required. This is an interesting direction to explore.

---

### Official Review · Reviewer_ie1L · 2022-07-10

**Rating:** 5
**Confidence:** 4
**Soundness:** 2 fair
**Presentation:** 2 fair
**Contribution:** 2 fair

**Summary:**

This paper investigates distributed sparse regression in high-dimensions to speed up its computation, which combines Nesterov’s proximal gradient with consensus and gradient-tracking mechanisms. This method can estimate locally the gradient of the empirical loss while enforcing agreement on the local estimates. Based on the basic assumptions, the proposed algorithm globally converges at linear rate. The iteration complexity scales as \sqrt{k} while the communications per iteration are at most \logm/(1-\rho). The experiments on real-world datasets and simulation studies verify the effectiveness of the proposed algorithms.

**Questions:**

This paper investigates acceleration for distributed sparse regression in high-dimensions. The paper can be improved from multiple points of view.
1)Presentation. The logic of this paper is quite hard to follow. The presentation could be improved.
2)Experiments. The experiment on more than one real-world data sets (only one in this paper). It would be better to do experiments with more real data.
3)Complexity of algorithm. It would have been nice to see the complexity (time and space) analysis of the proposed algorithm in theoretical analysis and experimental results.

**Limitations:**

This paper describes the scope of application of the proposed methods in detail, and does not involve negative social impact.

**Strengths And Weaknesses:**

This paper has the following contributions: This paper proposes distributed algorithm for high-dimensional estimation to reduce the computational requirements, which decentralizes Nesterov’s accelerated proximal gradient via consensus dynamics and a gradient tracking mechanism to track locally the gradient of f. In theoretical analysis, under standard notion of restricted strong convexity and smoothness of the empirical loss, the iterates generated by the proposed scheme converge at linear rate to a limit point. The numerical experiments on simulated and real data show the effectiveness of the proposed algorithm. The computational complexity is also an important part in algorithms. In this paper, it will be better to analyze the complexity (time, space, communications) of the algorithm from theoretical analysis and experiments.
The research field of this paper is significant. The organization and presentation of this paper can be further improved. This paper is quite hard to follow. There is no conclusion section and organization section.

---

> ### Author Response · Authors · 2022-08-02
> **Reply to Reviewer ie1L**
>
> We thank the Reviewer for her/his comments and suggestions for improving the presentation of the paper.
>   Our reply to her/his questions follow.
>
>
> $ \textbf{1. Presentation:}$ We apologize if the Reviewer found the presentation of the algorithm a bit dry to digest. Given the page constraints, we were forced to be concise and omit some intermediate details, referring to the literature, when possible. We understand that this makes the paper not self-contained. Following the Reviewer's suggestion, in the item 1 in the section of ``General comments'' at the beginning of this rebuttal, we have  provided more details on the main challenges one faces in designing distributed algorithms in high-dimension, the reason why one cannot use existing designs, and the rationale behind our design.    We hope that such clarifications will address the Reviewer's concerns/doubts. If not, we will be happy to elaborate more, based upon further Reviewer's requests. The Reviewer did not provide details on which specific parts of the algorithm description are not clear. We did our best to provide a generic, high-level clarification.
>
>
> $\textbf{2. Experiments:}$ We will add more experiments on real data in the revised version. For the time being we have made some preliminary results available at    https://anonymous.4open.science/r/NeurIPS2022-7EF0/NeurIPS_review_ie1L.pdf    for simulations on an additional data-set.  However, we would like to point out that the nature of this paper is quite theoretical and the numerical results we provided in the first place need to be indented more as a ``proof of concept'' than an extensive campaign of simulations benchmarking the proposed algorithms of different data sets. This is an important component of course, but it is not the main target of this study, given that all our results are formally proved.
>
> $\textbf{3. On the complexity of the algorithm:}$ We agree with the Reviewer that the complexity of the algorithm is an important component. In fact, in the paper we provided statistical and computational guarantees of the method, of the form that it is typical of such algorithms (see [1] as example in the centralized setting). Specifically, we  proved that the proposed method achieves an $\varepsilon$-statistically optimal solution in
>
> **(i)** a total number of iterations   of $$\mathcal{O}(\sqrt{\kappa}\log 1/\varepsilon);$$ **(ii)** a number of communications per iteration of $$\mathcal{O}\left(\frac{\log m \kappa}{1-\rho}\right);$$   **(iii)**  a total number of communications of  $$\widetilde{\mathcal{O}}\left(\sqrt{\kappa}\frac{\log m}{1-\rho}\log 1/\varepsilon\right);$$ and **(iv)**  the  number of local gradient evaluations to an $\varepsilon$ solution coincides with   that of the number of iterations. Therefore, the total time to reach an $\varepsilon$-solution is proportional to the number of iterations above,  with the constant that depends on the time to evaluate gradients, to  perform the projection onto the $\ell_1$ ball (which is of the order of $\mathcal{O}(d)$ operations), and perform vector-matrix multiplications, which are cpu/computer dependent.
>
> We point out  that the standard complexity in this contest (high-dimensional and distributed setting) is in the form (i)-(iii) above, which was discussed already in the paper. In particular, what is sensitive in high-dimension is the scaling of the above quantities with the ambient dimension $d$, as the typical regime of interest in high-dimension is the asymptotic $d/N\to \infty$. Therefore, it is desirable for a (distributed) algorithm in high-dimension to have complexity independent of $d$, which is a key future of the proposed method, along with the dependence on the square root of the restricted condition number $\kappa$ (proving acceleration). We do not know any other distributed algorithms enjoying  such desirable guarantees.
>
> The Reviewer may be interested  in the dependence of the  complexity results (i)-(iii) from the network topology and scaling with the   dimension. The table below shows the scaling of  the term  $(1-{\rho})^{-1}$ (appearing in the communication complexity, see (ii) and (iii) above) as a function of $m$ (and thus the communication complexity), for some representative topologies.
> $$\text{Line Graph}: \mathcal{O}(m^2)$$$$\text{2-d grid}: \mathcal{O}(m^2)$$ $$\text{Complete}: 1$$ $$\text{Star}: 1$$ $$\text{Erdos Renyi} \left(\text{link probability }p = \frac{\log m}{m} \right): \mathcal{O}(1)$$
> Therefore, depending on the specific topology, the total number of communications   ranges from polynomial scaling  in $m$ to a more favorably  logarithmic scaling.
>
>  $\textbf{4.}$ We added the conclusions, see the section of  ``General Comments'' at the beginning of the rebuttal.
>
> We hope that the above reply along with the discussion in the section of ``General Comments'' addressed the Reviewers' concerns. We will be happy to elaborate more, if the Reviewer is more specific on her/his requests.

---

> > ### Comment · Reviewer_ie1L · 2022-08-08
> > **Thank you for the responses**
> >
> > Thank you for your detailed response. My concerns are basically addressed. My score/rating has been updated.

---

### Official Review · Reviewer_GRMi · 2022-07-13

**Rating:** 7
**Confidence:** 4
**Soundness:** 3 good
**Presentation:** 3 good
**Contribution:** 3 good

**Summary:**

This manuscript studies the problem of sparse regression in a decentralized setting. Specifically, an accelerated algorithm is proposed, and its convergence and statistical properties are proven. Building upon recent advances in high-dimensional statistics, the authors take advance of the restricted strong convexity and restricted smoothness properties to develop their convergence analysis. Decomposable regularizers are also assumed. The authors show the performance of the proposed algorithm for the studied problems

**Questions:**

For what graph classes does the bound on \rho holds? More importantly, how does this bound scale with the number of agents? There is some dependency on m^2.5 which might imply that as the number of nodes increases, the graph needs to be more and more connected, which makes the proposed method impractical for relatively small graphs.

Can this method go beyond linear regression? Let's say strongly convex and smooth graphs?


**Limitations:**

No societal impact limitations.

**Strengths And Weaknesses:**

Strengths
- The manuscript clearly states the proposed problem and compares it with existing literature. It is clear that the proposed method extends the existing literature on the topic
- The manuscript is well written and easy to follow. The authors make an effort to explain the details in sufficient detail.

Weaknesses
- While there is sufficient contribution, many of the core technical ideas build upon [1] and classical approaches to decentralized optimization.
- "almost" strongly convex or smooth is strange. The way the authors handle inexactness and approximation terms like that causes confusion. For example, the authors mention trajectories (approximately) traveling in the "good" portion of the landscape. However, if the geometric properties of the function only hold in the appropriate subspaces, there is no reason to believe that they will hold in the approximate trajectories also.
- The authors mention and work with l0 in the sparse regression setting and numerics but it looks like l1 is used instead.
- The selected simulations do not clearly show the asymptotic behavior studied.

---

> ### Author Response · Authors · 2022-08-02
> **Reply to Reviewer GRMi - Part I**
>
> We thank the Reviewer for her/his positive comments on the assessment of the paper and her/his feedback. We hope that the reply below will address all her/his specific questions/comments. We will be happy to provide more details, if the Reviewer feels so. We number our answers according to the list of questions/comments under "Weaknesses'' and "Questions''.
>
> $\textbf{Reply to Weaknesses:}$
>
> $\textbf{1.}$  $\textbf{(i) Comparison with [1]:}$ While we agree with the Reviewer that our work has common elements with [1], they basically are the problem formulation (M-estimation problems), which we however studied in the distributed setting, and the technical assumptions on the centralized empirical loss (in particular, restricted strong convexity and smoothness), which are a consequence of the underlying studied statistical models. On the other hand, we believe that ''the core technical ideas'' behind the design of the algorithm and its analysis cannot be derived from [1].  $\textbf{First}$,  [1] deals with *centralized*, *non accelerated* projected gradient while we proposed an *accelerated and decentralized* scheme. Employing acceleration in high dimension is challenging. ``Classical'' acceleration of the proximal gradient methods do not lead to any acceleration in high dimension. We kindly ask the Reviewer to read item 1 of the "General comment'' section at the beginning of this rebuttal where we clarified   why standard accelerations and decentralizations of the proximal gradient method in [1]  yield unsatisfactory performance. A new design is needed, which departures from [1]. $\textbf{Second}$, the convergence analysis of the proposed method cannot build on that in [1]: a new, nontrivial  Lyapunov function is needed to account for acceleration and extra error terms due to the distributed nature of the scheme, coupled with a delicate analysis of those errors to provably achieve acceleration and steady-state errors of the order of the statistical precision.
>    $\textbf{(ii) Comparison with classical approaches in distributed optimization:}$ While the proposed distributed algorithm hinges on consensus and gradient tracking steps, the connection with the literature of distributed optimization ends there. There is in fact a substantial difference between the convergence analysis we put forth here and those in the literature of distributed optimization. As we discussed already in the paper (please see Sec. 1.1),  existing proofs in the literature of distributed optimization are of pure optimization type, lacking the statistical components. Roughly speaking, this means that  they $\textbf{(i)}$ would treat restricted strongly convex loss functions as convex ones, leading (assuming one can make those proofs work for the proposed algorithm) to  sublinear convergence rates,  let alone mentioning that the tuning of the parameters $\beta$ and $\alpha$ would be unclear, given that in high-dimension the condition number of the loss is no longer well defined; and $\textbf{(ii)}$ they would break down completely under the typical  high-dimensional scaling $d/n\to \infty$, as  the Lipschitz constant of the gradient scales as ${d}/{n}.$
>
> Hope this clarifies the novelty of our algorithm design and  the challenges we addressed in the analysis.
>
> $\textbf{2.}$   We kindly ask the Reviewer to refer to item 1 in the section of "General Comments'' at the beginning of this rebuttal, in particular to the discussion on the RSC condition (2). In a nutshell, under the RSC/RSM conditions,  enforcing feasibility of the iterate/momentum sequences is the key enabler to guarantee that the local gradients are always evaluated at points wherein the loss function exhibits "enough'' curvature (thus strong convexity, in the sense of (3)). Please let us know if the general comments at the beginning of the rebuttal provide a satisfactory answer.
>
> $\textbf{3.}$ We apologize but it  is unclear to us what the Reviewer means by  "the authors mention and work with the l0''. We do make the assumption that the ground truth is $s$-sparse, meaning  $||\theta^{\star}||_{\ell_0} = s$. However, we formulate the estimate of such a (exact) sparse parameter as a projected  linear regression problem, subject to  the $\ell_1$-ball constraint. As such, we never use the $\ell_0$ norm explicitly. Note that this is a standard practice in the context of linear sparse estimation.
>
> $\textbf{4.}$ We run more simulations as the Reviewer asked, considering problems with larger dimension $d$ and sample size.   Given the limited time to run new experiments, we were able to complete simulations with $d$ up to $10^6$, and ratios  $\frac{d}{N} = \{25,200,1667\}$. The results confirm that as long as $\frac{s * \log(d)}{d}$ remains invariant the achieved rate and statistical accuracy do as well, confirming our theoretical findings. Please, see the link https://anonymous.4open.science/r/NeurIPS2022-7EF0/NeurIPS_reviews_GRMi.pdf for the results and additional details.

---

> > ### Author Response · Authors · 2022-08-02
> > **Reply to Reviewer GRMi - Part II**
> >
> > $\textbf{Reply to ``Questions'':}$
> >
> > $\textbf{1.}$  We wish to clarify   that there is no limitation on the size of graph to satisfy the condition on the network connectivity $\rho$. More stringer requirements on $\rho$, not satisfied by the current graph and weight matrix $W$ in the first place, can be enforced just running     multiple steps of consensus and tracking updates in [S.1] per iteration $t$.  It is not difficult to check that, mathematically, $K$ consecutive rounds of consensus or gradient-tracking per iteration using each time weights  $W=[w_{ij}]$  (with associated $\rho=||W-J||$) correspond to use the same updates as in Eq. (12) and Eq. (13) in the paper  but with a new set of weights, say   $W'=[w_{ij}']_{ij=1}^m$, satisfying the property: $W'=W^K$. Denoting by  $\rho'= ||W'-J||$ the network connectivity of $W'$, the above relationship implies   $$\rho'=\rho^K.$$  Roughly speaking,   $K$ rounds of communications as above induce an ``effective'' network connectivity $\rho'$. Therefore, the condition in Eq. (18) in the paper (or similarly in Eq. (16)) on the network connectivity, $\rho \leq C m^{-2.5}\kappa^{-1}$, under $K$ rounds of communications, becomes
> > $$  \rho'\leq C m^{-2.5} \kappa^{-1}\quad \Rightarrow {\rho}^K\leq  C m^{-2.5} \kappa^{-1},$$ which is satisfied when the number of communications rounds $K$ is    $$ K \geq \frac{1}{1-{\rho}} \log \left(\frac{m^{2.5}\kappa}{C}\right).$$ This shows that, for any given graph and weight matrix $W$ (with associate, arbitrarily large  connectivity $\rho\in [0,1)$), the conditions on $\rho$ in Eq. (18) in the paper can *always* be satisfied by employing $K$ rounds of communications as above, resulting in a communication cost per iteration of $${\mathcal{O}}(\frac{\log (m\,\kappa)}{1-\rho}).$$  Note that this multiple rounds of communications are already incorporated in the total communication complexity $$\widetilde{\mathcal{O}}\left(\sqrt{\kappa}\frac{\log m}{1-\rho}\log1/\varepsilon\right),$$ as commented after Corollary 1 in the paper--see also Eq. (5) in the paper and comments therein.
> >
> >  As requested by the Reviewer, to provide some intuition on how the network size $m$ will affect  the total number of communications, the table below shows the scaling of  the term  $(1-{\rho})^{-1}$ (appearing in the communication complexity, see (ii) and (iii) below) as a function of $m$, for some representative topologies.
> > $$\text{Line Graph}: \mathcal{O}(m^2)$$$$\text{2-d grid}: \mathcal{O}(m^2)$$ $$\text{Complete}: 1$$ $$\text{Star}: 1$$ $$\text{Erdos Renyi} \left(\text{link probability }p = \frac{\log m}{m} \right): \mathcal{O}(1)$$
> > Therefore, depending on the specific topology, the total number of communications   ranges from polynomial scaling  in $m$ to a more favorably  logarithmic scaling.
> >
> >
> >
> > $\textbf{2.}$  The proposed method is applicable to other M-estimation problems. For instance, we can cover all the M-estimator studied in [1] (but here in a distributed setting and employing acceleration). Referring to other estimators, the algorithm is technically applicable; however,  the convergence analysis  would require some adjustment. For instance, we are currently investigating statistical and computational guarantees for *generalized* linear models.

---

> > > ### Author Response · Authors · 2022-08-09
> > > **Any further question?**
> > >
> > > Can you please let us know if our answers are satisfactory and address your concerns? There are only few hours left to keep up with the discussion and update the score to reflect the final outcome of the discussion.
> > >
> > > Thanks
> > >
> > > The authors

---

> > > > ### Comment · Reviewer_GRMi · 2022-08-10
> > > > **Satisfactory answers**
> > > >
> > > > Yes, I will increase my score to 7.

---

> > > > > ### Author Response · Authors · 2022-08-10
> > > > > **Thanks**
> > > > >
> > > > > Thanks for finding the time to check our replies.

---

### Official Review · Reviewer_UFEg · 2022-07-18

**Rating:** 7
**Confidence:** 2
**Soundness:** 3 good
**Presentation:** 2 fair
**Contribution:** 3 good

**Summary:**

The paper proposes a method for accelerating distributed high-dimensional regression under sparsity constraints. To do so the authors adapt Nesterov's proximal gradient method to the setting by combining it with a consensus step based on the connectivity of the graph. Experiments on synthetic data and a couple of public datasets demonstrates superiority validate the analytical claims.

**Questions:**

Please address the points under Weaknesses above (provide intuition for (13); explain what multiple rounds of communication/iteration means, how it would help, and what effect it would have on guarantees; provide a conclusion section)

**Strengths And Weaknesses:**

Strengths:

1. The paper clearly improves over prior work in theory and in practice.

2. The application of Nesterov's method to this setting is clear and intuitive and overall the authors demonstrate its impact on multiple problem settings.

Weaknesses:

1. The explanation of the method is a bit dense and may not make complete sense to readers unfamiliar with all the technical prerequisites. Specifically, the logic behind equation (13) was not clear to me. It appears to rely on prior work ([7],[30],[19]) but as it is the key step in the algorithm, I would strongly recommend explaining it in a self-contained manner to make the picture clearer.

2. The authors acknowledge that when the graph is not connected enough (for e.g. when condition (18) is not satisfied) then multiple rounds of communication/iteration may be needed to facilitate convergence. It is not clear what that exactly means. I am assuming it means multiple rounds of S.1 (consensus and gradient tracking). However I do not see clearly how that will overcome the lack of connectivity. This is another place where some intuition would be helpful. Also won't it affect the statistical and computational guarantees in (5) since now each iteration will have higher cost due to multiple rounds of communication?

3. The paper ends abruptly with no conclusion section or a discussion of limitations/future work.

---

> ### Author Response · Authors · 2022-08-02
> **Reply to Reviewer UFEg**
>
> We thank the Reviewer for her/his positive feedback, which will help to improve the presentation of the revised paper. Our reply to her/his questions under the ``Weaknesses'' follows (using the same numbering).
>
> $\textbf{1.}$  We apologize for the dry description of the  algorithm design and limited insights. This was because of space limit. We elaborate next on the  intuition   behind  the gradient tracking mechanism  (13) (we use the same notation as in the paper).  The goal of the tracking variable $y_i^t$ is to approximate the gradient of the average loss $F$. In fact, under the initialization $y_i^0=\nabla f_i(z^0_i)$, we can write
> $$ \small  \overline{y}^t\triangleq \frac{1}{m}\sum_{i=1}^m y_i^t = \frac{1}{m} \sum_{i=1}^m y_i^{t-1} +  \frac{1}{m} \sum_{i=1}^m \nabla f_i(z_i^{t}) - \frac{1}{m} \sum_{i=1}^m \nabla f_i(z_i^{t-1}) $$ $$\quad =\cdots =\frac{1}{m} \sum_{i=1}^m \nabla f_i(z_i^{t}) + \frac{1}{m}\sum_{i=1}^m y_i^{0} - \frac{1}{m} \sum_{i=1}^m \nabla f_i(z_i^{0})=\frac{1}{m} \sum_{i=1}^m \nabla f_i(z_i^{t}),
> $$
> where the first equality follows from the tracking update in Eq. (13) in the paper   and the column-stochasticity of the weight matrix $W$ ($1^\top W = 1^\top$); and in the second row we applied the equality in the first raw telescopically while using the initialization $y_i^0=\nabla f_i(z^0_i)$. In parallel the update on the z-variable as in Eq. (12) in the paper aims at enforcing consensus among the local $z_i^t$ (and by-product on the y-variables) which asymptotically for $t\to \infty$ leads to $$z_i^t \approx \overline{z}^t\triangleq \frac{1}{m}\sum_{i=1}^m z_i^t\quad \text{and}\quad  y_i^t \approx \overline{y}^t = \frac{1}{m} \sum_{i=1}^m \nabla f_i(z_i^{t}) \approx \frac{1}{m} \sum_{i=1}^m \nabla f_i(\overline{z}^t)=\nabla F(\overline{z}^t).$$
>
> Therefore, gradient tracking and consensus in tandem  yields an improving estimate of $\nabla F$ by the tracking variables.
>
> $\textbf{2.}$ The Reviewer is correct:  multiple rounds of communications mean performing multiple steps of consensus and tracking updates in [S.1] for  each iteration $t$. It is not difficult to check that, mathematically, $K$ consecutive rounds of consensus or gradient-tracking per iteration using each time weights  $W=[w_{ij}]$  (with associated $\rho=||W-J||$) correspond to use the same updates as in Eq. (12) and Eq. (13) in the paper  but with a new set of weights, say   $W'=[w_{ij}']_{ij=1}^m$, satisfying the property: $W'=W^K$. Denoting by  $\rho'= ||W'-J||$ the network connectivity of $W'$, the above relationship implies   $$\rho'=\rho^K.$$  Roughly speaking,   $K$ rounds of communications as above induce an ``effective'' network connectivity $\rho'$. Therefore, the condition in Eq. (18) in the paper (or similarly in Eq. (16)) on the network connectivity, $\rho \leq C m^{-2.5}\kappa^{-1}$, under $K$ rounds of communications, becomes
> $$  \rho'\leq C m^{-2.5} \kappa^{-1}\quad \Rightarrow {\rho}^K\leq  C m^{-2.5} \kappa^{-1},$$ which is satisfied when the number of communications rounds $K$ is    $$ K \geq \frac{1}{1-{\rho}} \log \left(\frac{m^{2.5}\kappa}{C}\right).$$ This shows that, for any given graph and weight matrix $W$ (with associate, arbitrarily large  connectivity $\rho\in [0,1)$), the conditions on $\rho$ in Eq. (18) in the paper can *always* be satisfied by employing $K$ rounds of communications as above, resulting in a communication cost per iteration of $${\mathcal{O}}(\frac{\log (m\,\kappa)}{1-\rho}).$$  Note that this multiple rounds of communications are already incorporated in the total communication complexity $$\widetilde{\mathcal{O}}\left(\sqrt{\kappa}\frac{\log m}{1-\rho}\log1/\varepsilon\right),$$ as commented after Corollary 1 in the paper.
>
>
>
> $\textbf{3.}$   We thank the Reviewer for suggesting to include some conclusions. We reported in the *item 2* of the general comments at the beginning of this rebuttal the conclusions we will add to the paper.
>
> We hope that the above reply addresses all the questions of the Reviewers. Please let us know if more details are needed and we will be happy to elaborate accordingly.

---

> > ### Comment · Reviewer_UFEg · 2022-08-08
> > **Re**
> >
> > Thank you for addressing my concerns. The explanation of the gradient and consensus tracking step is much clearer now and should be included in the final version of the paper. Regarding point 2 above, can you clarify how $\rho' = \rho^K$? Specifically, how does multiplying $W$ by itself $K$ times lead to $||W' - J||$ becoming equal to $||W-J||^K$?

---

> > > ### Author Response · Authors · 2022-08-08
> > > **Reply to the Reviewer**
> > >
> > > Thanks for checking our reply, we are glad they you find our answers satisfactory. We will add the description of the gradient tracking in the revision  of the paper along with the conclusions.
> > >
> > > Referring to your last question, the aforementioned equality comes from the following fact $$W^K-J=(W-J)^K,$$
> > >
> > > where the equality follows from $WJ=JW=J$ due to the doubly stochasticity of $W$.
> > >
> > > We hope that this addresses all the questions/concerns you had. Please let us know otherwise.

---

> > > > ### Comment · Reviewer_UFEg · 2022-08-09
> > > > **Re**
> > > >
> > > > Yes, my concern is addressed. Thank you. I have increased my score to 7.

---

### Author Response · Authors · 2022-08-02
**General Comments- Part I**

We thank the Reviewers for their  insightful comments, which will help to improve the presentation of our results. Before addressing individual questions, we wish to provide some clarifications on the genesis of the algorithm and contrast with existing   methods in the literature (non applicable in high dimension) as well as address some common questions/requests raised by the Reviewers.


$\textbf{1. About the assessment of the proposed algorithms:}$ From the assessment of the paper on the algorithm design as summarized by some  Reviewers, we realized that our presentation might have created some misunderstanding  about the connection of the proposed (centralized and distributed) accelerated  schemes and  the widely used classical Nesterov's proximal gradient and its decentralization, DPAG, in [33]. In a nutshell,  there is no direct connection: the proposed accelerated methods, centralized (see Eq. (9)  in the paper) and distributed (Algorithm 1 in the paper), cannot be derived by   the classical  Nesterov's proximal gradient. Acceleration in the (distributed) high-dimensional setting needs to be
rethought–new designs and convergence analyses are needed. Next, we elaborate on this and provide some insight on the genesis of the new algorithms.

$\quad\textbf{(i) Centralized method:}$ The  Nesterov's (accelerated) proximal gradient  reads [21]:
\begin{equation}   \theta^{t+1} = \text{argmin}_{x: \mathcal{R}(x) \leq r} \langle \nabla f(x^t), x - x^t\rangle + \frac{L_g}{2}||x - x^t||^2 , \quad  x^{t+1} = \theta^{t+1} +  \frac{ \sqrt{\kappa_g} - 1}{ \sqrt{\kappa_g} +1} (\theta^{t+1} - \theta^t).\qquad (1)
\end{equation}
 To provably achieve acceleration, (1)    requires the objective loss $f$ to be $\mu_g$-strongly convex and $L_g$-smooth *on the entire space $\mathbb{R}^d$* (in fact, in (1), $\kappa_g=L_g/\mu_g$).  Therefore, this scheme is *not* applicable to the   high-dimensional setting considered in our paper where the loss $f$ is   strongly convex and smooth only in a restricted set of directions (which would imply $\mu_g=0$). This is also confirmed by the numerical results in Fig. 1 in the paper where the decentralization of (1) performs poorly in high-dimension [and so does the centralized instance  (1) above].

 This motivates a new design that explicitly accounts for the Restricted Strong Convexity (RSC) and Restricted Smoothness (RSM) of the loss function. We elaborate on the RSC only. The RSC reads (see Eq. (6) in the paper):
   \begin{equation}
       f(x) - f(y) - \langle \nabla f(y), x - y \rangle  \geq \frac{\mu}{4}||x-y||^2_2 - \frac{\tau_\mu}{2}\mathcal{R}^2(x - y),\quad \forall x,y\in \Omega.\qquad (2)
   \end{equation}
This condition imposes a curvature on the loss $f$ only along the directions $x$, $y\in \Omega$ such that the right hand side of (2) is positive, i.e., \begin{equation}
       \frac{\mathcal{R}^2(x-y)}{||x-y||^2}<\frac{\mu}{2\tau_{\mu}}.\qquad (3)\end{equation}
This guides the new algorithm design: the trajectory $x^t$ (used to compute the gradient of $f$) should travel in the region where $x^t-\theta^\star$ satisfies (3), so that the algorithm experiences roughly strongly convex losses. These are sparse directions (``small'' values of $\mathcal{R}(x^t-\theta^\star)$; note that  $\theta^{\star}$ is sparse).
Unfortunately, this is not the case for the trajectory of the Nesterov's proximal gradient in  (1): $x^t$ is not feasible (i.e., does not satisfy $\mathcal{R}(x^t)\leq r$); hence, (3) is generally violated along $x^t-\theta^\star$, having the algorithm experiencing a nonstrongly convex loss. This also explains the poor behaviour of such a scheme in high-dimension.

The proposed algorithm in Eq. (9) in the paper overcomes this issue, generating by design trajectories that are feasible for $\mathcal{R}$ (thus compliant with (3). This together with a careful control of the consensus and tracking errors yield for the first time   in high dimension the typical acceleration of strongly convex losses.

$\quad \textbf{(ii) On the  decentralized accelerated algorithm:}$  The same comments as above can be made for the existing decentralization of the classical Nesterov's proximal acceleration, the DGPA in [33], which as such inherits the same issue of  (1) above. This is also supported by the simulations in Fig. 1, showing the poor behaviour of DGPA in high dimension. The decentralization of the   accelerated centralized algorithm in Eq. (9) of the paper is done to preserve, in the distributed settings,  the key feature discussed above enabling acceleration, leading to Algorithm 1 in the paper.

We hope that the above discussion provides some insight on the  genesis of the new algorithm in Eq. (9) and its distributed instance, Algorithm 1,  as well as how the proposed novel approach copes with the main issues of  the classical  Nesterov's proximal acceleration   in high-dimension.  We would like to stress one more time that all this was unknown before this work.

---

> ### Author Response · Authors · 2022-08-02
> **General Comments - Part II**
>
> $\textbf{2. Conclusions:}$ Reviewers $\textbf{UFEg}$ and $\textbf{ie1L}$ are asking to add some conclusions in the paper;  in the revised version, we will add the following.
>
> We   proposed the first  decentralized accelerated algorithm  to solve high dimensional estimation problems over mesh networks  whose empirical losses   are neither strongly convex nor Lipschitz smooth globally. To employ acceleration in this setting, the design hinges on   careful considerations regarding the directions traversed by the schemes, enforcing  sparsity of the iterate and momentum sequences.   Global convergence to statistically optimal solutions is proved  at liner rate, proportional to $\sqrt{\kappa}$, with a communication cost per iteration of $\widetilde{\mathcal{O}}(\log m/(1-\rho))$. It is unclear whether this communication cost is improvable, for instance, whether the log-dependence on the number of agents can be alleviated or so the transmission of all $d$ gradient/iterate components. This is left to future investigations.

---

### Meta-Review · Area_Chair_jq2P · 2022-08-27

**Recommendation:** Accept
**Confidence:** Less certain

**Metareview:**

The paper provides novel guarantees for the well-studied distributed sparse regression problem. Their theoretical results improve upon the state of the art and extend to settings that many previous results could not handle. From a technical perspective, their result builds upon previous frameworks, but also requires a number new, novel ideas. The paper does have some downsides; as mentioned previously, some of their ideas do build quite strongly off of previous work, and the presentation of the paper is quite dense, as several reviewers noted. However, the consensus of the reviewers overall is that the technical contribution of the paper is above the bar for acceptance, and would be of interest to the distributed optimization community.

**Award:**

No

---

### Decision · Program_Chairs · 2022-09-14

Accept